# Site-specific phosphorylation and caspase cleavage of GFAP are new markers of Alexander disease severity

Rachel A Battaglia[1], Adriana S Beltran[2,3], Samed Delic[1,4], Raluca Dumitru[3], Jasmine A Robinson[1], Parijat Kabiraj[1†], Laura E Herring[2], Victoria J Madden[5], Namritha Ravinder[6], Erik Willems[6], Rhonda A Newman[6], Roy A Quinlan[4], James E Goldman[7], Ming-Der Perng[8], Masaki Inagaki[9], Natasha T Snider[1*]

[1]Department of Cell Biology and Physiology, University of North Carolina, Chapel Hill, United States; [2]Department of Pharmacology, University of North Carolina, Chapel Hill, United States; [3]Human Pluripotent Stem Cell Core, University of North Carolina, Chapel Hill, United States; [4]Department of Biosciences, University of Durham, Durham, United Kingdom; [5]Department of Pathology, University of North Carolina, Chapel Hill, United States; [6]Thermo Fisher Scientific, Carlsbad, United States; [7]Department of Pathology, Columbia University, New York, United States; [8]Institute of Molecular Medicine, National Tsing Hua University, Hsinchu, Taiwan, Republic of China; [9]Department of Physiology, Mie University Graduate School of Medicine, Mie, Japan

*For correspondence:
ntsnider@med.unc.edu

Present address: †Department of Neurology, Mayo clinic, Rochester, United States

**Abstract** Alexander disease (AxD) is a fatal neurodegenerative disorder caused by mutations in glial fibrillary acidic protein (GFAP), which supports the structural integrity of astrocytes. Over 70 GFAP missense mutations cause AxD, but the mechanism linking different mutations to disease-relevant phenotypes remains unknown. We used AxD patient brain tissue and induced pluripotent stem cell (iPSC)-derived astrocytes to investigate the hypothesis that AxD-causing mutations perturb key post-translational modifications (PTMs) on GFAP. Our findings reveal selective phosphorylation of GFAP-Ser13 in patients who died young, independently of the mutation they carried. AxD iPSC-astrocytes accumulated pSer13-GFAP in cytoplasmic aggregates within deep nuclear invaginations, resembling the hallmark Rosenthal fibers observed in vivo. Ser13 phosphorylation facilitated GFAP aggregation and was associated with increased GFAP proteolysis by caspase-6. Furthermore, caspase-6 was selectively expressed in young AxD patients, and correlated with the presence of cleaved GFAP. We reveal a novel PTM signature linking different GFAP mutations in infantile AxD.

## Introduction

Alexander disease (AxD) is a rare and invariably fatal neurological disorder that affects primarily infants and small children, but can also manifest later in life (*Alexander, 1949*; *Sosunov et al., 2018*; *Messing, 2018*). Autosomal dominant gain-of-function mutations in *GFAP*, which encodes glial fibrillary acidic protein (GFAP), cause AxD (*Messing, 2018*; *Brenner et al., 2001*). GFAP is the major component of the intermediate filament (IF) cytoskeleton in astrocytes (*Hol and Pekny, 2015*). The accumulation and incorporation of mutant GFAP within cytoplasmic aggregates called Rosenthal fibers (RFs), causes reactive gliosis, leading to secondary injury to neurons and non-neuronal cells (*Olabarria and Goldman, 2017*; *Wang et al., 2015*; *Li et al., 2018*; *Jones et al., 2018*). Silencing *GFAP* via antisense oligonucleotide intervention in vivo eliminates RFs, reverses the stress

**eLife digest** Alexander disease is a rare and fatal neurodegenerative condition. It is caused by mutations in a gene that is crucial for the structure of astrocytes, a type of brain cell whose role is to support neurons. The gene codes for a protein called GFAP, which is made almost exclusively in astrocytes. In Alexander disease, mutated versions of the gene make GFAP collect in disordered clumps or aggregates, which interfere with the astrocytes' normal activities.

All Alexander disease patients develop GFAP aggregates, but the type of mutation they have in the gene for GFAP does not predict how their illness will progress. The age of onset of disease, for example, can vary between less than one year old to more than 70 years old. Battaglia et al. sought to understand how GFAP aggregates form in the cells of Alexander disease patients. One way that GFAP can be altered in the cell is by a process called phosphorylation. Enzymes called kinases add phosphate groups to GFAP, which can regulate the protein's activity, stability and interactions with other proteins.

Battaglia et al. found high levels of phosphorylation at one specific site in the GFAP protein in people who had very early onset of Alexander disease. This phosphorylation was not related to any particular mutation in the gene for GFAP. An added phosphate group at this location in the protein made GFAP more likely to be broken into two pieces by an enzyme called caspase-6. One of the breakdown products is already known to play a role in aggregation. Young patients with Alexander disease had high levels of GFAP breakdown products and caspase-6. The phosphorylated protein and this enzyme were found to accumulate in astrocyte aggregates.

The findings provide a basis for investigating new strategies to treat Alexander disease that target phosphorylation – as removing or preventing the addition of phosphate groups can be done with drugs. But before exploring how to do this, it will be necessary to find out which enzyme is responsible for phosphorylating GFAP at this particular position. These studies may also give a broader understanding of other GFAP-like proteins (called intermediate filament proteins), which are involved in over 70 human diseases.

responses in astrocytes and other cell types, and improves the clinical phenotype in a mouse model of AxD (*Hagemann et al., 2018*). While the utility of GFAP as a key therapeutic target in AxD is clear, the molecular mechanisms for how AxD-associated GFAP missense mutations (affecting over 70 different residues on GFAP) lead to defective GFAP proteostasis are not well understood. Deciphering these mechanisms may yield novel interventions, not only for AxD patients, but also for patients with other diseases where IF proteostasis is severely compromised.

Normal functioning IFs are stress-bearing structures that organize the cytoplasmic space, scaffold organelles, and orchestrate numerous signaling pathways. In contrast, dysfunctional IFs directly cause or predispose to over 70 tissue-specific or systemic diseases, including neuropathies, myopathies, skin fragility, metabolic dysfunctions, and premature aging (*Omary, 2009*; www.interfil.org). Disease-associated IF proteins share two key molecular features: abnormal post-translational modifications (PTMs) (*Snider and Omary, 2014*) and pathologic aggregation. The GFAP-rich RF aggregates that are hallmarks of AxD astrocytes bear strong similarities to pathologic aggregates of other IFs, including epidermal keratins (*Coulombe et al., 1991*), simple epithelial keratins (*Nakamichi et al., 2005*), desmin (*Dalakas et al., 2000*), vimentin (*Müller et al., 2009*), neurofilaments (*Zhai et al., 2007*) and the nuclear lamins (*Goldman et al., 2004*). There are unique advantages to studying IF proteostasis mechanisms in the context of GFAP because of its restricted cellular expression, homopolymeric assembly mechanism, and because GFAP is the sole genetic cause of AxD as a direct result of its toxic gain-of-function accumulation and aggregation.

Like all IF proteins, GFAP contains three functional domains: amino-terminal 'head' domain, central α-helical 'rod' domain and carboxy-terminal 'tail' domain (*Eriksson et al., 2009*). The globular head domain is essential for IF assembly and disassembly, which are regulated by various PTMs, in particular phosphorylation (*Omary et al., 2006*). It was shown previously that phosphorylation of multiple sites in the head domain of GFAP (Thr-7, Ser-8, Ser-13, Ser-17 and Ser-34) regulates filament disassembly during mitosis and GFAP turnover in non-mitotic cells (*Inagaki et al., 1990*; *Takemura et al., 2002a*; *Inagaki et al., 1994*; *Inagaki et al., 1996*). Additionally, phosphorylation of

GFAP has been observed after various injuries of the central nervous system (CNS) including kainic acid-induced seizures, cold-injury, and hypoxic-ischemic models, where phosphorylated GFAP is expressed in reactive astrocytes (*Valentim et al., 1999*; *Takemura et al., 2002b*; *Sullivan et al., 2012*). These observations reveal that phosphorylation of GFAP is important for re-organization of the astrocyte IF cytoskeleton and plasticity in response to injury. However, it is not clear if, and how, abnormal GFAP phosphorylation compromises proteostasis and contributes to AxD pathogenesis.

Here, we identified a critical phosphorylation site in the GFAP head domain that is selectively and strongly upregulated in the brain tissues of AxD patients who died very young, independently of the position of the disease mutation that they carried. Further, we show that this site-specific phosphorylation promotes GFAP aggregation and is a marker of perinuclear GFAP aggregates associated with deep nuclear invaginations in AxD patient astrocytes, but not in isogenic control astrocytes. Finally, we demonstrate a correlation between site-specific GFAP phosphorylation and caspase cleavage in cells and in post-mortem brain tissue from AxD patients. Although our study does not establish a causal relationship between GFAP phosphorylation and caspase cleavage, we show that caspase-6 is a new marker for the most severe form of human AxD.

Collectively, our results reveal a new PTM signature that is associated with defective GFAP proteostasis in the most severe form of AxD. Future interventional studies targeting these PTMs will determine whether they contribute to, or are the consequence of, disease severity.

## Results

### Phosphorylation of Ser13 on GFAP is a marker of the most aggressive form of AxD

IFs undergo protein synthesis-independent turnover and re-organization to meet cellular demands (*Robert et al., 2016*). PTMs are key in that process, as they regulate filament polymerization and depolymerization, protein-protein interactions, and oligomerization properties of IF proteins (*Snider and Omary, 2014*). Of all known PTMs that regulate IFs, phosphorylation is the most ubiquitous and can facilitate or antagonize other types of PTMs via complex cross-talk mechanisms (*Omary et al., 2006*). We hypothesized that AxD-associated GFAP missense mutations (*Figure 1A*) promote GFAP accumulation and aggregation by dysregulating site-specific phosphorylation. We extracted GFAP from post-mortem brain cortex tissue of 13 AxD patients, representing 10 different mutations (*Supplementary file 1*) and three non-AxD controls (*Supplementary file 2*). GFAP from the insoluble high salt extracts (HSEs), prepared according to the procedure described in *Figure 1— figure supplement 1*, was used in phospho-proteomic analysis, revealing 12 unique phosphorylation sites on GFAP in AxD (*Figure 1B–C*). While the AxD-specific phospho-peptides localized to all three functional domains of GFAP (head, rod, tail), the most abundantly phosphorylated residue was a conserved serine (Ser13) in the head domain (*Figure 1C–D*).

Strikingly, we found that the pSer13-GFAP peptide was selectively elevated in the cortex tissue from AxD patients who died very young (median age at death = 1.7 years; range 0.5–14 years) (*Figure 2A*). Overall, we did not observe significant phosphorylation of GFAP in the control subjects (*Figure 2—source data 1*), or in AxD patients who lived 27–50 years (median age at death = 38 years). Further, immunoblot analysis using a phospho-specific antibody (KT13) (*Sekimata et al., 1996*) against pSer13-GFAP validated the mass spectrometry results in the AxD patients (*Figure 2B–C*). Although there was one notable outlier in each age group (*Figure 2B* lanes 3 and 11), our results suggest that pSer13-GFAP is primarily associated with the more aggressive, infantile form of AxD. Furthermore, the differences in phosphorylation were not a result of age, since pSer13 GFAP was generally not present in the brain lysates from non-AxD control subjects, regardless of age (*Figure 2D*).

### Phospho-mimic mutation at Ser13 promotes GFAP aggregation

To determine the functional significance of pSer13 on GFAP filament organization, we analyzed the filament properties of non-phosphorylatable (S13A) and phospho-mimic (S13D and S13E) GFAP mutants. We optimized a transient over-expression system in the SW13 vimentin-negative adreno-carcinoma cells (SW13vim-) for this assay, which resulted in primarily filamentous WT GFAP and insoluble aggregated forms of common AxD mutants of GFAP (*Figure 3—figure supplement 1*).

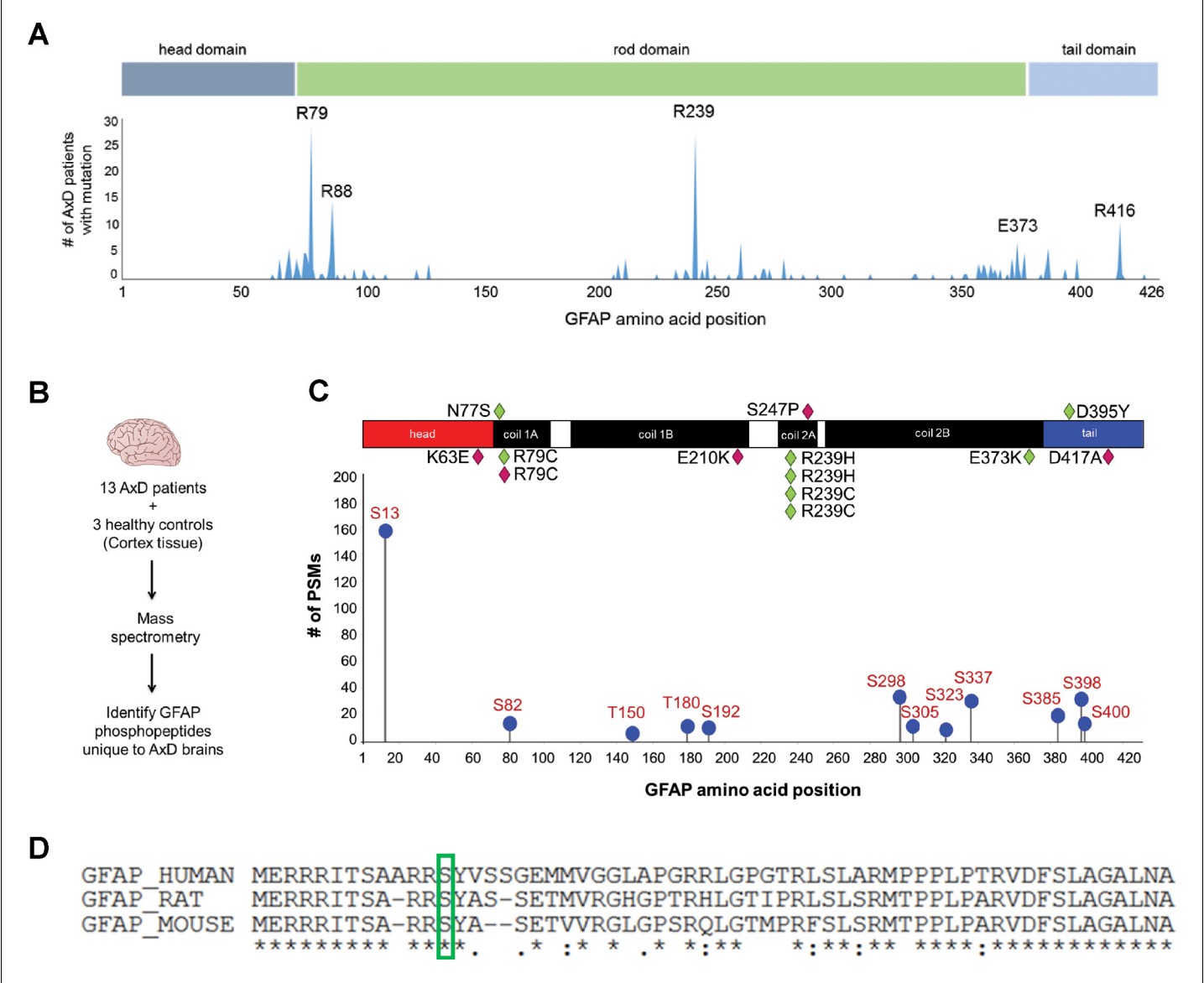

**Figure 1.** GFAP is phosphorylated on head domain Ser13 in human AxD brain. (**A**) Schematic displays the frequency and location of AxD patient GFAP mutations. (**B**) Method used to identify GFAP phospho-peptides. (**C**) Graph of AxD-specific GFAP phospho-peptides identified by mass spectrometry and type/position of patient mutations. PSM = peptide spectrum match. Green diamonds represent GFAP mutations in young patients (median age at death = 1.7 years; range 0.5–14 years) and pink diamonds represent older patients (median age at death = 38 years; range 27–50 years). (**D**) Amino acid conservation at the N-terminus of human, rat and mouse GFAP. The green box indicates the serine corresponding to human Ser13, which is conserved in rat and mouse.

The online version of this article includes the following figure supplement(s) for figure 1:

**Figure supplement 1.** Preparation of brain high salt extracts (HSE) for mass spectrometry analysis of GFAP.

Compared to wild-type (WT) GFAP, the S13D and S13E mutants assembled primarily into large aggregates, similar to the most common AxD-associated mutant R79H-GFAP (*Figure 3A–B*). S13A formed mostly filaments, although they appeared shorter compared to WT GFAP. To determine if the phospho-mimic mutation directly promotes aggregation, we compared the assembly properties of purified WT, S13A and S13D GFAP (*Figure 3C*). Consistent with the phenotype observed in the transfected cells, the S13A mutant formed abnormally short filaments in vitro. In contrast, S13D was completely incapable of filament assembly, forming globular structures that were homogeneous in size and not aggregation-prone. Our results with the phospho-deficient and phospho-mimic mutants

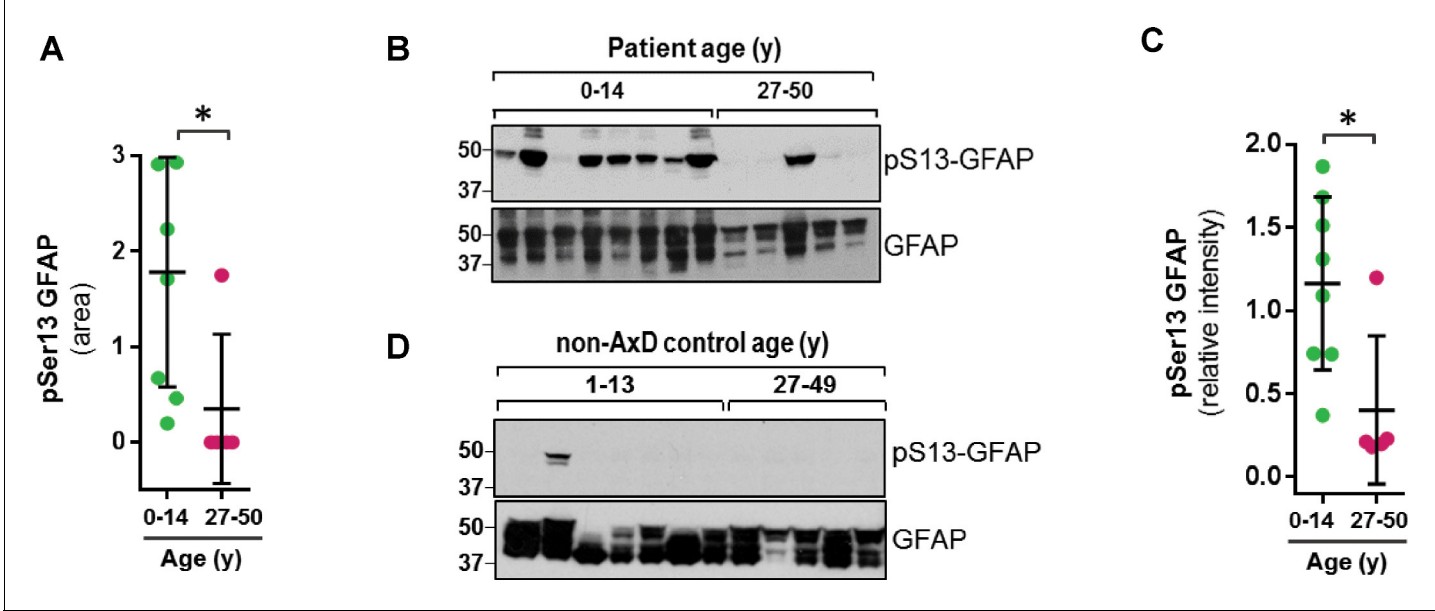

**Figure 2.** GFAP is phosphorylated on head domain Ser13 primarily in AxD brain from young patients. (A) Quantification of pSer13-GFAP abundance by mass spectrometry in young (green) vs. old (pink) AxD patients (*=p < 0.05 unpaired t-test). (B) Validation of pSer13-GFAP by western blot of HSE from AxD patients, using a phospho-specific antibody to pSer13-GFAP. The order of samples, by AxD donor ID number, is: 1482, 1070, 885, 5488, 1161, 2768, 338, 613, 5377, 5517, M3596, 5109, and 4858 (listed in *Supplementary file 1*). (C) Quantification of the relative intensity of pSer13-GFAP on western blot in young (green) and old (pink) AxD patients (*=p < 0.05 unpaired t-test). Signal intensity was normalized to total GFAP in each sample. (D) Western blot of pSer13-GFAP in non-AxD control brain lysates of different ages. The order of samples, by donor ID number, is: 1547, 5941, 103, 1791, 1670, 4898, 1706, 1711, 1011, 632, 4640, and 4915 (listed in *Supplementary file 2*).

The online version of this article includes the following source data for figure 2:

**Source data 1.** Raw data from mass spectrometry PTM profiling of GFAP extracted from AxD and control human brain.

reveal that S13 is a key site that regulates the assembly properties of GFAP and that its phosphorylation status may modulate the dynamics between filaments and aggregates.

## Generation of AxD induced pluripotent stem cells (iPSCs) and isogenic controls

In order to explore the function of this phosphorylation event in a disease-relevant system, we used an in vitro human astrocyte model of AxD. We generated iPSCs using fibroblasts from a young AxD patient and characterized their pluripotency by immunofluorescence staining (*Figure 4A*). Karyotype analysis showed that there were no chromosomal abnormalities due to the reprogramming process (*Figure 4—figure supplement 1*). To generate isogenic control cells, we corrected the heterozygous point mutation in *GFAP* (c.715C > T, p.R239C) using CRISPR/Cas9 mediated gene editing (*Figure 4B*). Representative chromatograms are shown for the original patient cells and the isogenic controls (*Figure 4C*). We also isolated 'CRISPR control' clones, which were edited on the wild-type *GFAP* allele, thereby retaining the AxD-causing mutation and serving as an additional disease control for the gene editing procedure. Similar to the original patient cells, the edited cells were karyotyped and characterized for pluripotency (*Figure 4—figure supplement 1*). We confirmed that there were no off-target effects due to the editing procedure (*Supplementary file 3*).

## GFAP accumulation and perinuclear aggregation into RF-like structures in AxD iPSC-astrocytes

AxD, CRISPR control, and isogenic control iPSCs were differentiated to astrocytes (iPSC-astrocytes) via neural progenitor cells (NPCs), as described in the Materials and methods and shown schematically in *Figure 4D*. After 54 days in culture, iPSC-astrocytes express classical astrocyte markers (*Zhang et al., 2014*), including alcohol dehydrogenase 1 family member L1 (ALDH1L1), solute carrier family 1 member 3 (SLC1A3), excitatory amino acid transporter 2 (EAAT2), Connexin 43 and GFAP

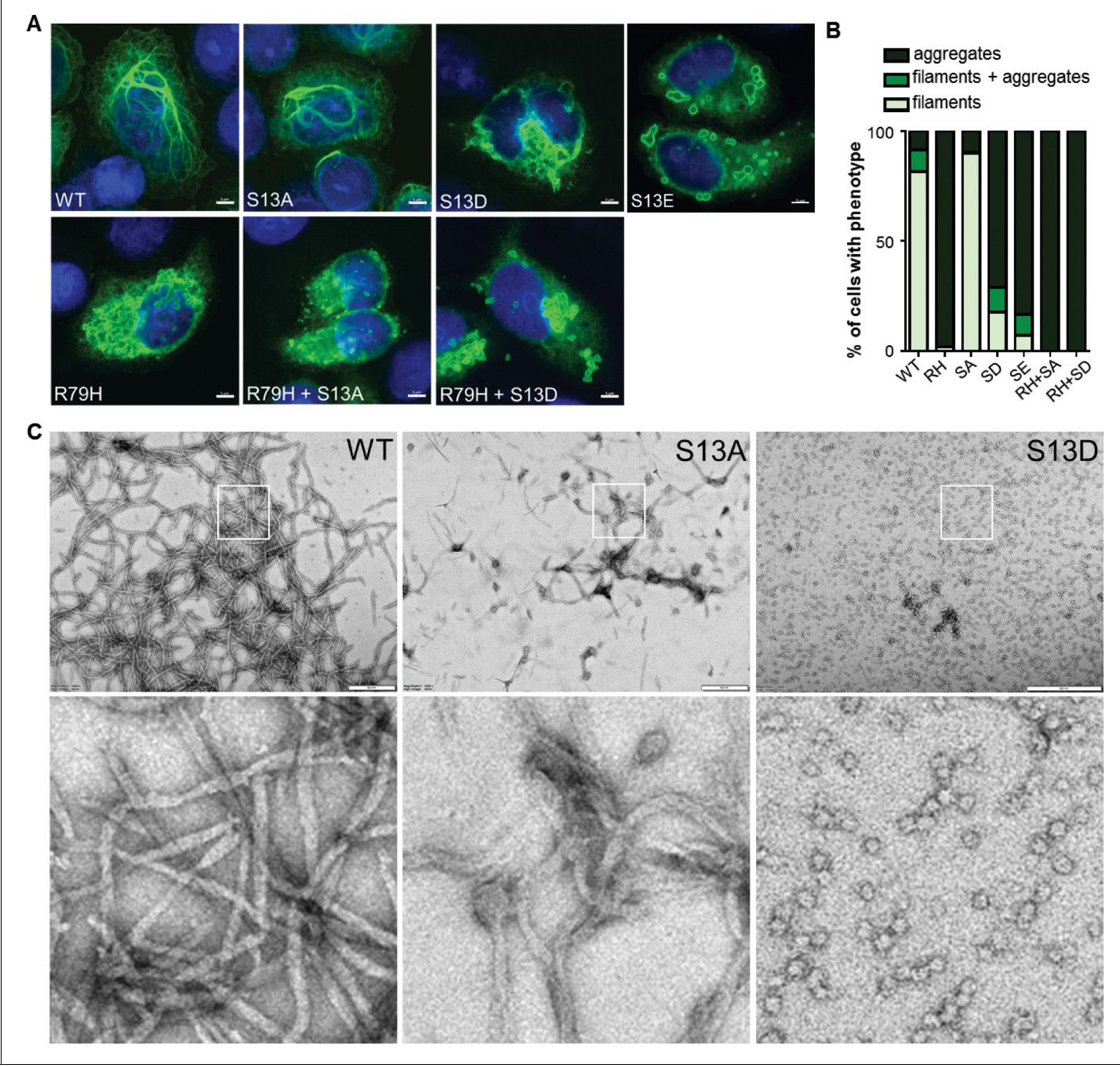

**Figure 3.** Effect of phospho-deficient and phospho-mimic S13 substitutions on GFAP filament assembly in cells and in vitro. (**A**) Representative images of immunofluorescence staining of DNA (blue) and GFAP (green) in SW13vim- cells transfected with wild-type GFAP (WT), R79H mutant GFAP (R79H), non-phosphorylatable GFAP (S13A), and phospho-mimic GFAP (S13D and S13E) as single or double mutations, as noted in the images. Scale bar = 5 µm. (**B**) Quantification of percentage of cells containing GFAP filaments, aggregates or both (n = 41–103 cells per condition). RH = R79H; SA = S13A; SD = S13D; SE = S13E. (**C**) Electron micrographs showing the filament properties of in vitro assembled GFAP (WT, S13A and S13D). Bottom three panels represent magnified areas marked by the white boxes in the top panels. Scale bars = 500 nm.

The online version of this article includes the following figure supplement(s) for figure 3:

**Figure supplement 1.** Optimization of transient expression for WT and AxD-associated GFAP mutant proteins in SW13vim- cells.

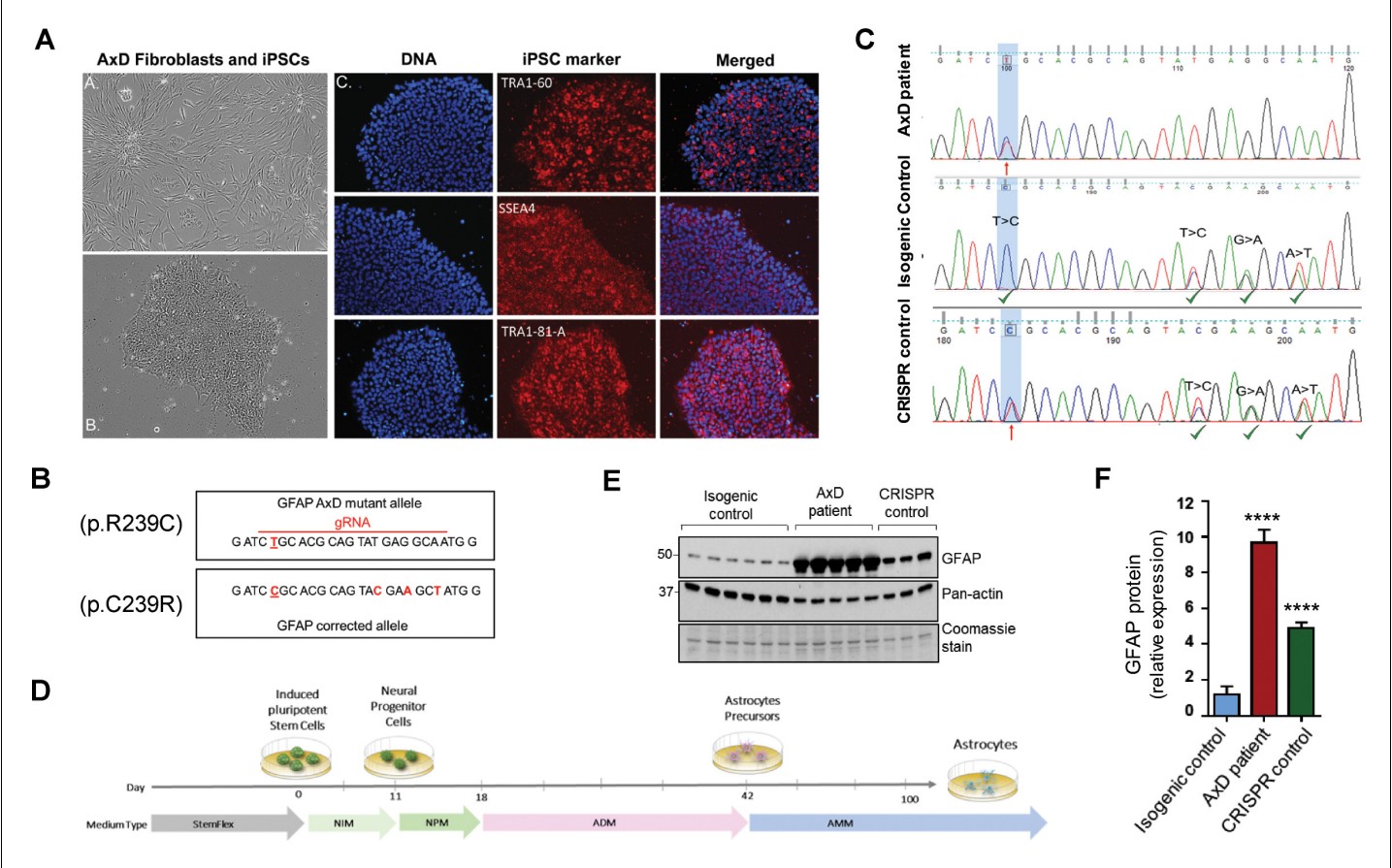

**Figure 4.** Generation and characterization of AxD patient iPSC-astrocytes and isogenic controls. (**A**) Characterization of iPSC pluripotency. Bright field images of AxD patient fibroblasts (top left) and iPSCs (bottom left). Immunofluorescence staining for pluripotency markers in AxD iPSCs. (**B**) GFAP sequence for the AxD mutant allele and the corrected allele. Differences between the sequences are indicated by red text. The AxD-causing mutation is underlined, and all other changes are silent mutations. The area of gRNA recognition is indicated by the red line. (**C**) Chromatograms showing AxD heterozygous mutation in the original patient cells (top), correction of the mutant allele in the isogenic control (middle) and correction of the wild-type allele in the CRISPR control (bottom). Red arrows denote presence of the disease mutation and green check mark denote genetic correction and presence of silent mutations. (**D**) Schematic representation of astrocyte differentiation protocol. NIM, neural induction medium; NPM; neural progenitor medium; ADM, astrocyte differentiation medium; AMM; astrocyte maturation medium. (**E**) Immunoblot of GFAP in iPSC-astrocytes. Pan-actin blot and Coomassie stain serve as loading controls. (**F**) Quantification of band intensities for GFAP from panel E. ****$p < 0.0001$ compared to isogenic control; one-way ANOVA.

The online version of this article includes the following figure supplement(s) for figure 4:

**Figure supplement 1.** Characterization of pluripotency in AxD and isogenic control iPSCs.

**Figure supplement 2.** Characterization of astrocyte differentiation.

(*Figure 4—figure supplement 2*). To assess if our model recapitulated key features of AxD, we analyzed total GFAP expression in the iPSC-astrocytes by immunoblot, and found that GFAP levels were significantly higher in the cells that carried the heterozygous *GFAP* point mutation (AxD patient and CRISPR control lines) relative to the isogenic controls (*Figure 4E–F*). This is consistent with in vivo observations of GFAP levels in AxD patients (*Jany et al., 2015*) and mouse models (*Jany et al., 2013*). In addition, high-molecular-mass (hmm) GFAP oligomers were present in the AxD iPSC-astrocytes, similar to what we observed when we ectopically expressed the R239C-GFAP mutant (*Figure 5A*). Finally, we observed by immunofluorescence staining that the AxD mutant iPSC-astrocytes formed both GFAP filaments and perinuclear aggregates (*Figure 5B*), whereas the isogenic control iPSC-astrocytes formed only GFAP filaments (*Figure 5C*). In vivo, GFAP antibodies stain the periphery, while DAPI stains the core of RFs (*Der Perng et al., 2006*; *Sosunov et al., 2017*).

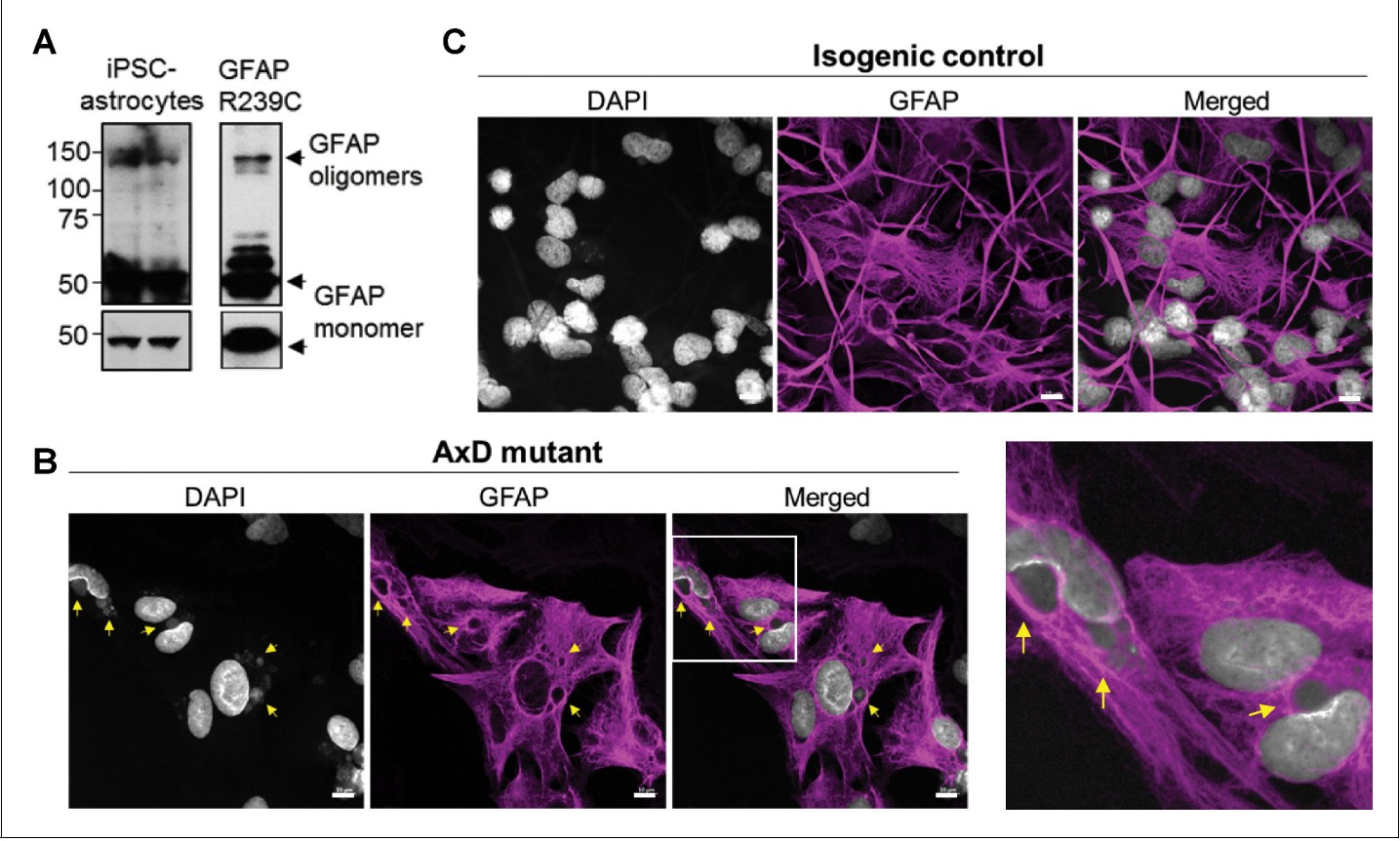

**Figure 5.** Oligomerization and perinuclear aggregation of GFAP in AxD iPSC-astrocytes. (**A**) GFAP blot of AxD iPSC-astrocytes (left) and SW13vim-cells transfected with R239C mutant GFAP (right) reveals GFAP monomer and high-molecular-mass GFAP oligomers. Immunoblots on the bottom are of the same membranes at lower exposure. (**B**) Immunofluorescence staining for GFAP (magenta) and DAPI (white) in AxD iPSC-astrocytes reveals presence of perinuclear GFAP aggregates, marked by the yellow arrows. Scale bars = 10 µm. Boxed area in the merged image is shown by the enlarged image on the right. (**C**) Immunofluorescence staining for GFAP (magenta) and DAPI (white) in isogenic control iPSC-astrocytes. Scale bars = 10 µm.

The in vitro-derived AxD iPSC-astrocytes displayed similar characteristics, with RF-like perinuclear aggregates staining positively for GFAP at their periphery and DAPI in the center (*Figure 5B*).

## pSer13-GFAP marks the core of perinuclear GFAP aggregates localized within deep nuclear invaginations

Next, we determined if pSer13-GFAP was present in the AxD iPSC-astrocytes, similar to what we observed in the human brain tissues. As shown in *Figure 6A*, pSer13-GFAP signal was detected strongly within the core of the perinuclear GFAP aggregates of AxD iPSC-astrocytes. Somewhat surprisingly, we also observed pSer13-GFAP signal in the isogenic control cells, possibly triggered by the in vitro culture conditions. Nevertheless, unlike AxD astrocytes, in the isogenic control astrocytes pSer13-GFAP organization was filamentous and paralleled that of total GFAP. Therefore, the in vitro iPSC-astrocyte model revealed that, only in the presence of the AxD disease mutation, pSer13-GFAP is incorporated within the core of perinuclear inclusions. While in all AxD cells pSer13 signal was detected in the aggregates, we also observed cells with pSer13-positive diffuse cytoplasmic staining and filaments, likely reflecting different states of the GFAP network (*Figure 6—figure supplement 1*). Furthermore, the pSer13-positive GFAP aggregates appeared adjacent to prominent nuclear invaginations (*Figure 6A*). Nuclear deformations, similar to what we observed in the AxD iPSC-astrocytes, are also present in RF-bearing astrocytes in AxD human brain (*Sosunov et al., 2017*). To determine whether the perinuclear aggregates compromised the nuclear envelope, we

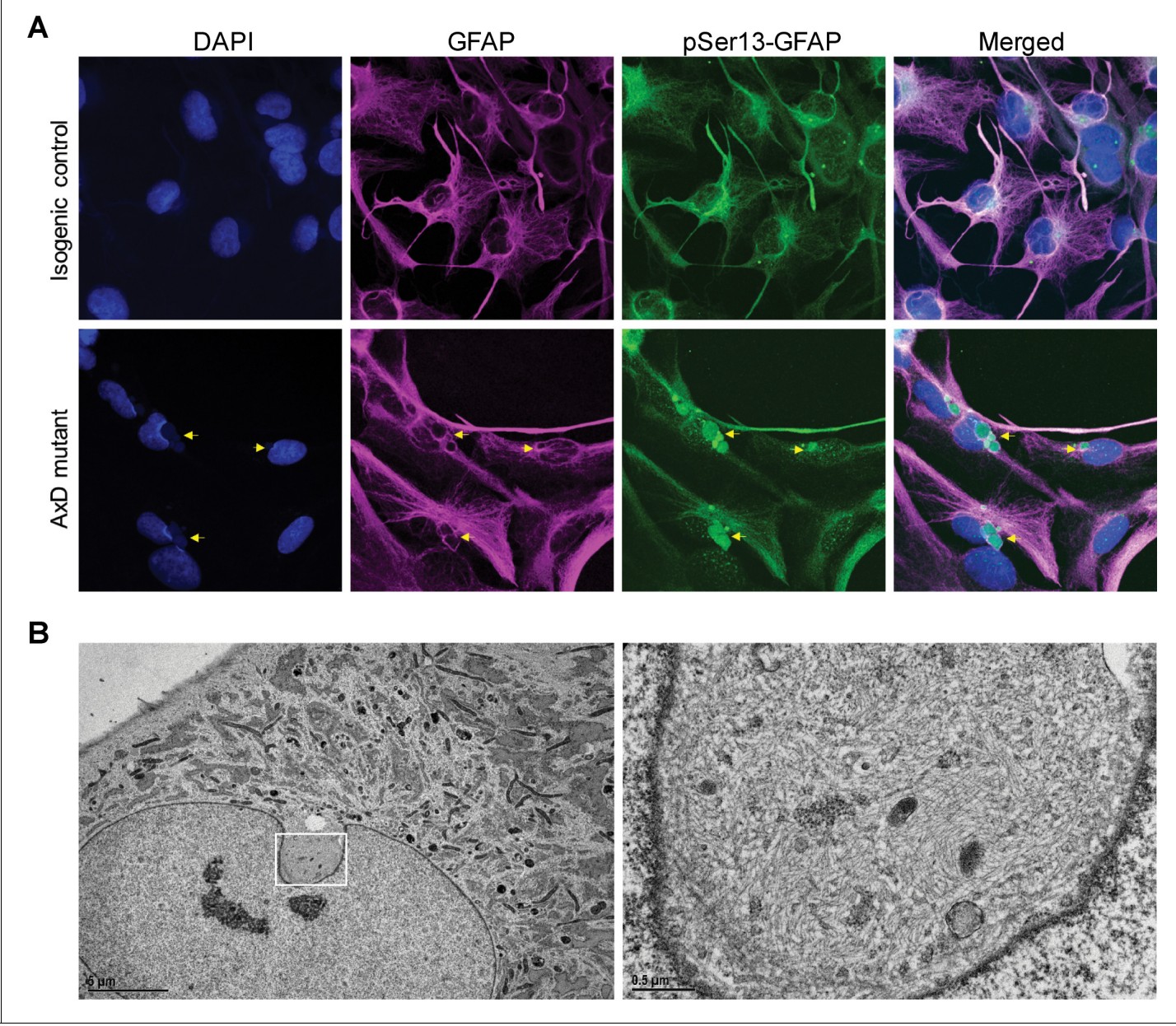

**Figure 6.** pSer13 marks perinuclear accumulation of GFAP within nuclear invaginations in AxD iPSC-astrocytes. (**A**) Immunofluorescence staining of total GFAP (magenta), pSer13-GFAP (green) and DAPI (blue) in isogenic control (top panels) and AxD mutant (bottom panels) iPSC-astrocytes. Perinuclear GFAP aggregates are indicated by the yellow arrows. Scale bars = 10 μm. (**B**) Electron microscopy images of AxD patient iPSC-astrocytes revealing large, juxtanuclear fibrous bundles (boxed area on left), shown at higher magnification on the right. Scale bar = 5 μm (left) and 0.5 μm (right). The online version of this article includes the following figure supplement(s) for figure 6:

**Figure supplement 1.** Three types of staining pattern observed with the pSer13 GFAP antibody in AxD iPSC-astrocytes.

examined the AxD iPSC-astrocytes by electron microscopy. While we observed filamentous bundles on the cytoplasmic side of the nuclear invaginations, the nuclear envelope appeared intact (*Figure 6B*). Thus, pSer13-GFAP marks cytoplasmic GFAP aggregates adjacent to nuclear invaginations. It should be noted that the perinuclear aggregates containing disorganized GFAP filaments

are not identical to the electron-dense RFs that are seen in post-mortem patient brain, but that they may reflect an intermediate state of GFAP accumulation.

## Phosphorylation at Ser13 promotes caspase-mediated cleavage of GFAP

To understand the mechanism for how GFAP phosphorylation may promote GFAP aggregation, we conducted a biochemical analysis of the S13A, S13D and S13E GFAP mutants. In line with our immunofluorescence result (*Figure 3A*), we observed an increase in high-molecular-mass ~100 kDa GFAP oligomer in the phospho-mimic mutant by immunoblot analysis (*Figure 7A*). However, more strikingly, we observed increased levels of a cleaved GFAP fragment (24 kDa) in S13D and S13E, which was significantly lower in WT- and S13A-GFAP (*Figure 7A–B*). Cleavage of GFAP by caspase-6 in vitro generates two fragments of 24 and 26 kDa size (*Chen et al., 2013*). The 24 kDa C-terminal fragment is recognized by the monoclonal GA5 antibody, (*Chen et al., 2013*) which was used here. Therefore, we tested the effect of a peptide inhibitor of caspase-6 (Ac-VEID-CHO), and found that it significantly reduced the amount of cleaved S13D-GFAP (*Figure 7C–D*). Furthermore, we observed augmented cleavage of S13D-GFAP when combined with an AxD-causing mutation (S13D/R79H double mutant), and this was also blocked by the caspase-6 inhibitor (*Figure 7C–D*). Further analysis of the AxD mutant R79H in the transfection system revealed phosphorylation not only at S13, but also at nearby Y14, S16, and S17 (*Figure 7—figure supplement 1* and *Figure 7—source data 1*). Of note, mutagenesis of S16 and S17 to non-phosphorylatable alanines reduced both the cleavage and oligomerization of R79H (*Figure 7—figure supplement 1*). Phospho-motif analysis revealed that S13, S16 and S17 are part of a segment in the GFAP head domain that is a potential target for several kinases (*Supplementary file 4*). Candidate kinases include casein kinase 2 (CK2), protein kinase A (PKA), PKC, MAP kinase activated protein kinase 2 (MAPKAP2), and glycogen synthase kinase 3 (GSK3). These data suggest that phosphorylation of Ser13 (and nearby S16/17) may promote caspase-6-mediated cleavage of GFAP in the context of AxD mutations. In line with that, we observed increased levels of cleaved GFAP (upon normalization for total GFAP) in the AxD iPSC-astrocytes compared to isogenic control astrocytes (*Figure 7E*), along with intense caspase-6 staining within perinuclear GFAP aggregates in AxD iPSC-astrocytes, but not isogenic control astrocytes (*Figure 7F*).

## Interference with GFAP cleavage by caspase-6 partially reduces aggregation of the phospho-mimic mutant S13D

To determine how blocking GFAP cleavage affects aggregation, we performed site-directed mutagenesis to block cleavage of GFAP at Asp225. As shown in *Figure 8A–B*, the D225E mutation reduced cleavage of S13D GFAP by >90%. This resulted in partial rescue of filament structure in S13D, although the D225E mutation on its own caused significant filament bundling and perinuclear structures that resembled large aggregates (*Figure 8C–D*). We also tested the effect of the caspase-6 inhibitor Ac-VEID-CHO, and found that it reduced both the size of the S13D aggregates (*Figure 8E*) and the presence of ~100 kDa hmm GFAP oligomers (*Figure 8F–G*). However, similar to the mutagenesis experiment, filament bundles were observed in WT and S13D GFAP treated with Ac-VEID-CHO, suggesting that caspase-6 regulates both aggregation and normal GFAP filament reorganization.

## Caspase-6 expression and GFAP cleavage are upregulated in AxD patients

Caspase-6 is not expressed highly in the normal human brain, especially after birth (*Godefroy et al., 2013*). Therefore, we wanted to examine its expression in the context of AxD. Using immunoblot analysis of total brain lysates, we found that caspase-6 is expressed in the brain tissue from all 8 AxD patients who died very young, but is essentially undetectable in the patients who survived longer (*Figure 9A*). To ensure caspase-6 expression is not simply more abundant in young individuals, we compared brain lysates from young and old AxD patients to non-AxD control brains from age-matched individuals, and observed a significant increase in caspase-6 expression selectively in young AxD patients, but not in the other groups (*Figure 9B–C*).

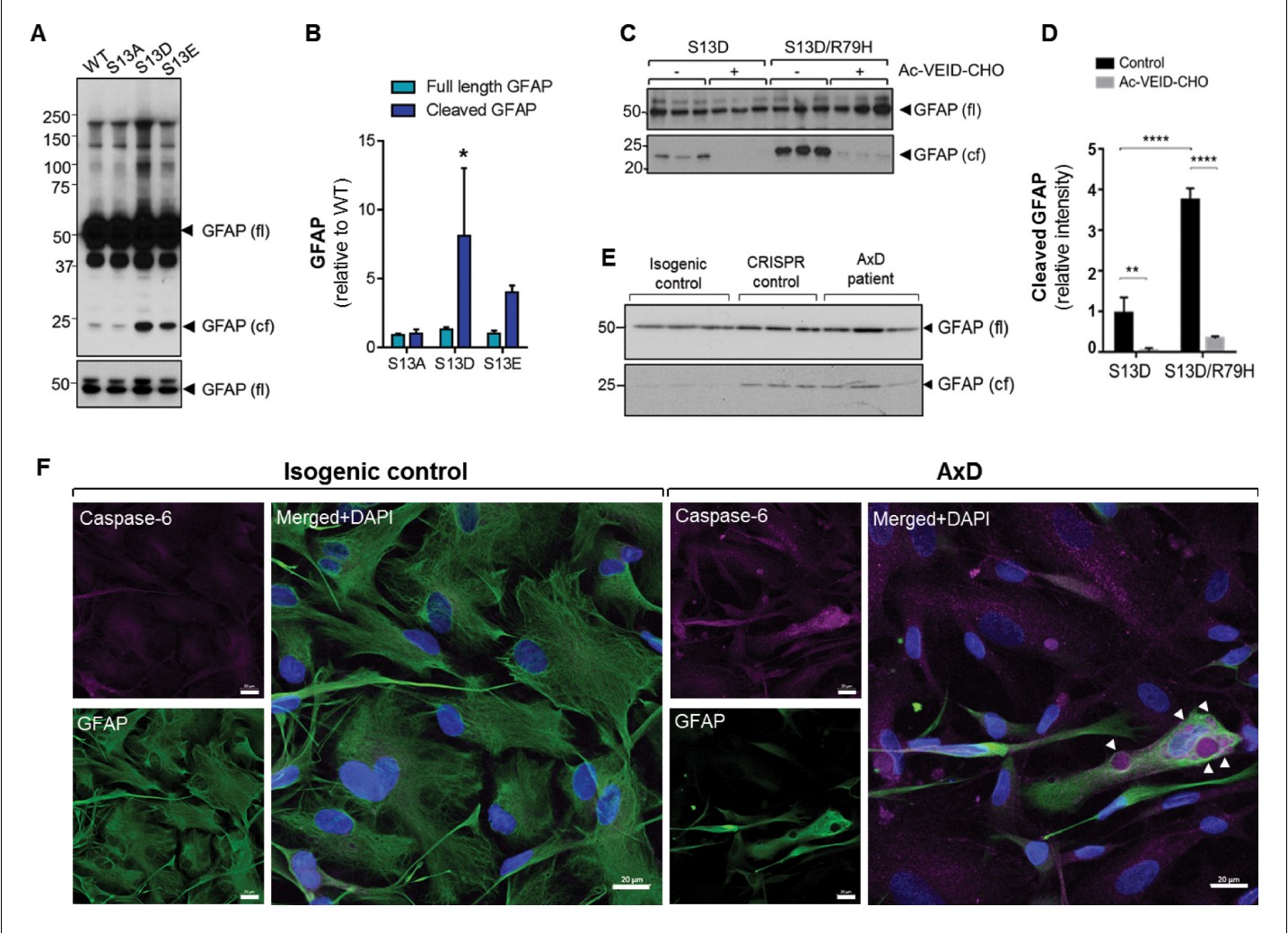

**Figure 7.** Phosphorylation of Ser13 on GFAP promotes caspase-6 cleavage of GFAP. (A) GFAP blot of SW13vim- cells transfected with vector, WT, S13A, S13D and S13E - GFAP. Full-length (fl) and cleaved fragment (cf) of GFAP are indicated by arrows. Immunoblot on the bottom shows GFAP monomer (fl) from the same membrane at a lower exposure. (B) Quantification of panel A by densitometry shows cleaved and full-length GFAP in phospho-mutants relative to WT GFAP (mean ± SD from three independent experiments; *p<0.05 two-way ANOVA). (C) GFAP blot in SW13vim- cells transfected with either S13D or S13D/R79H double mutant GFAP and treated for 48 hr with a caspase-6 inhibitor (Ac-VEID-CHO). (D) Quantification of GFAP bands in panel C by densitometry (mean ± SD from three biological replicates; **p<0.01; ****p<0.0001 two-way ANOVA). (E) Immunoblot for GFAP monomer (fl) and cleaved fragment (cf) in isogenic control and AxD iPSC-astrocytes. Different amounts of total protein were loaded to normalize GFAP monomer levels. (F) Immunofluorescence staining of caspase-6 (magenta), GFAP (green) and DAPI (blue) in human AxD and isogenic control iPSC-astrocytes showing caspase-6 co-localization within GFAP aggregates in the AxD cells, indicated by the arrowheads. Scale bars = 20 μm.

The online version of this article includes the following source data and figure supplement(s) for figure 7:

**Source data 1.** Raw data from mass spectrometry PTM profiling of GFAP R79H extracted from transfected SW13vim-cells.

**Figure supplement 1.** Analysis of major sites of phosphorylation on R79H GFAP expressed in SW13 vim- cells.

Next we asked whether AxD patients, particularly young AxD patients that exhibit more pSer13-GFAP and caspase-6 expression, also displayed increased GFAP cleavage. To determine the extent of caspase-6-cleaved GFAP in AxD patient brains, we utilized an antibody that specifically recognizes N-terminally caspase-6-cleaved GFAP (D225) (*Chen et al., 2013*). We detected cleaved GFAP in extracts from AxD patient brains, and we observed a significant increase in the amount of D225 signal in young AxD patients, which paralleled the increased pSer13 signal in these samples (*Figure 9D–E*). In agreement with the biochemical evidence, brain tissues from young AxD patients stained intensely for cleaved GFAP, while the signal was significantly weaker in AxD patients who were older (*Figure 9F*; *Figure 9—figure supplement 1*). The signal was particularly strong around

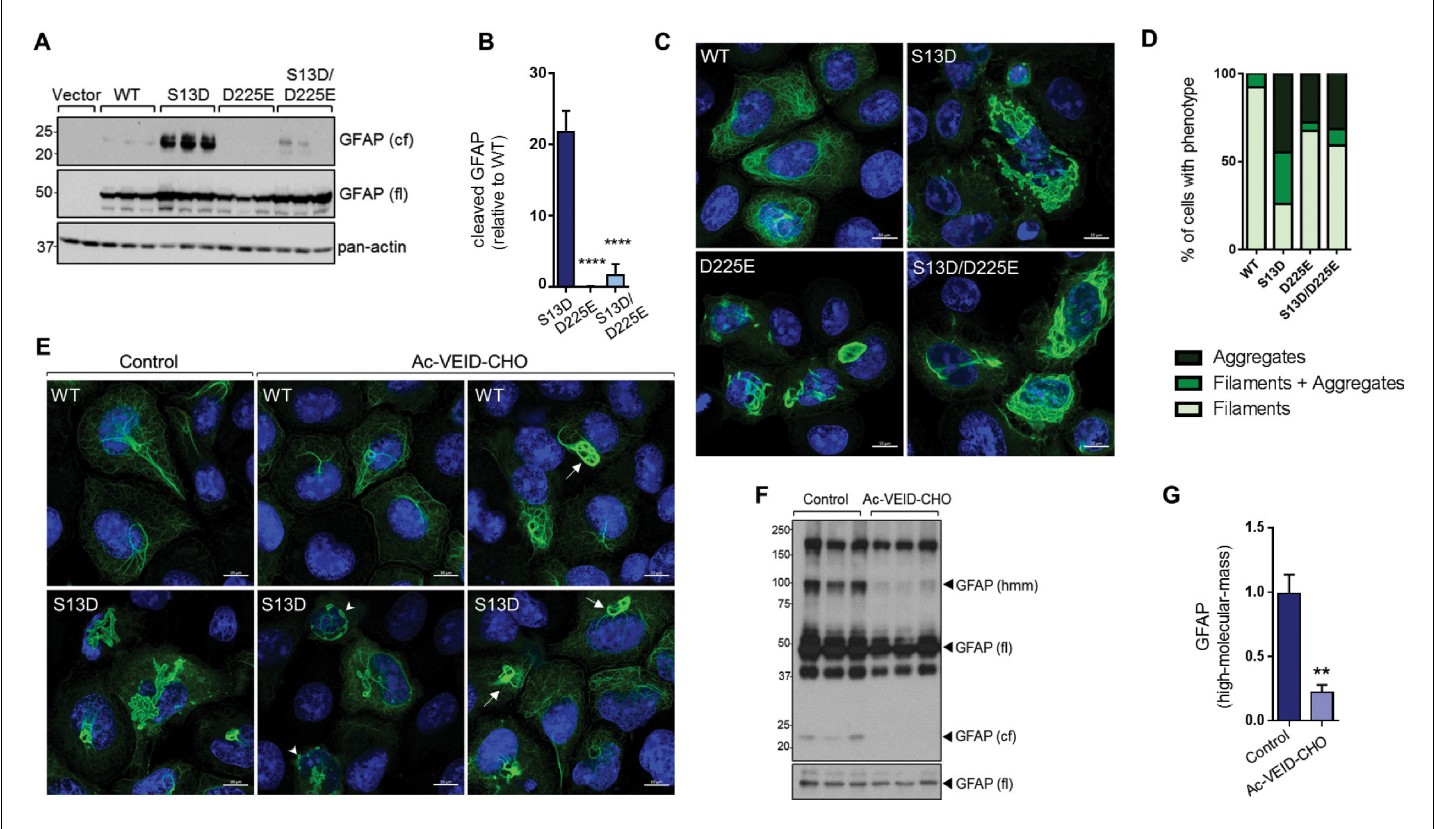

**Figure 8.** Inhibition of GFAP cleavage by caspase-6 partially alleviates aggregation due to S13D phospho-mimic mutation. (**A**) Western blot of GFAP total cell lysates from SW13vim- cells transfected with empty vector control, WT, S13D, D225E, and double S13D/D225E mutants. Shown are GFAP cleaved fragment (cf), full-length (fl) monomer and pan-actin (loading control). (**B**) Quantification of the abundance of cleaved GFAP in the three mutants shown in panel A relative to WT GFAP (mean ± SD from three biological replicates; ****p<0.0001 compared to S13D; one-way ANOVA). (**C**) Representative images of immunofluorescence staining of DNA (blue) and GFAP (green) in SW13vim- cells transfected with wild-type GFAP (WT), phospho-mimic GFAP (S13D), and non-cleavable GFAP (D225) as single or double mutations, as noted in the images. Scale bar = 10 μm. (**D**) Quantification of percentage of cells containing GFAP filaments, aggregates or both (n = 76–85 cells per condition). (**E**) Representative images of immunofluorescence staining of DNA (blue) and GFAP (green) in SW13vim- cells transfected with wild-type GFAP (WT) or phospho-mimic GFAP (S13D) and treated with vehicle (control) or the caspase-inhibitor Ac-VEID-CHO (10 μM, 48 hr). (**F**) Western blot analysis of SW13vim- total lysates transfected with S13D GFAP and treated with vehicle (control) or caspase-6 inhibitor Ac-VEID-CHO (10 μM, 24 hr), showing the 24 kDa caspase-cleaved fragment (cf), 50 kDa full-length (fl), and high-molecular-mass ~100 kDa GFAP. (**G**) Quantification of the relative abundance of hmm GFAP in control and Ac-VEID-CHO – treated cells. n = 3; **p<0.01; unpaired t-test.

perinuclear areas and surrounded circular structures that stained positive for DAPI (*Figure 9F*, bottom panels), similar to what we observed in the AxD iPSC-astrocytes. Thus, our results show that caspase-6 expression in AxD patient brain tissue parallels the presence of cleaved GFAP, and both are selectively and significantly elevated in patients who succumbed to the disease very early in life.

## Discussion

Our study reveals that missense mutations, affecting discrete domains on the GFAP molecule, share a common PTM signature that is associated with compromised GFAP proteostasis in the severe form of AxD. Using patient brain tissue and human iPSC-derived AxD astrocytes, we show that head domain phosphorylation promotes defective filament assembly and perinuclear accumulation and incorporation of mutant GFAP within nuclear invaginations. By taking an unbiased mass spectrometry proteomic approach, we were able to identify GFAP phospho-peptides that were selectively

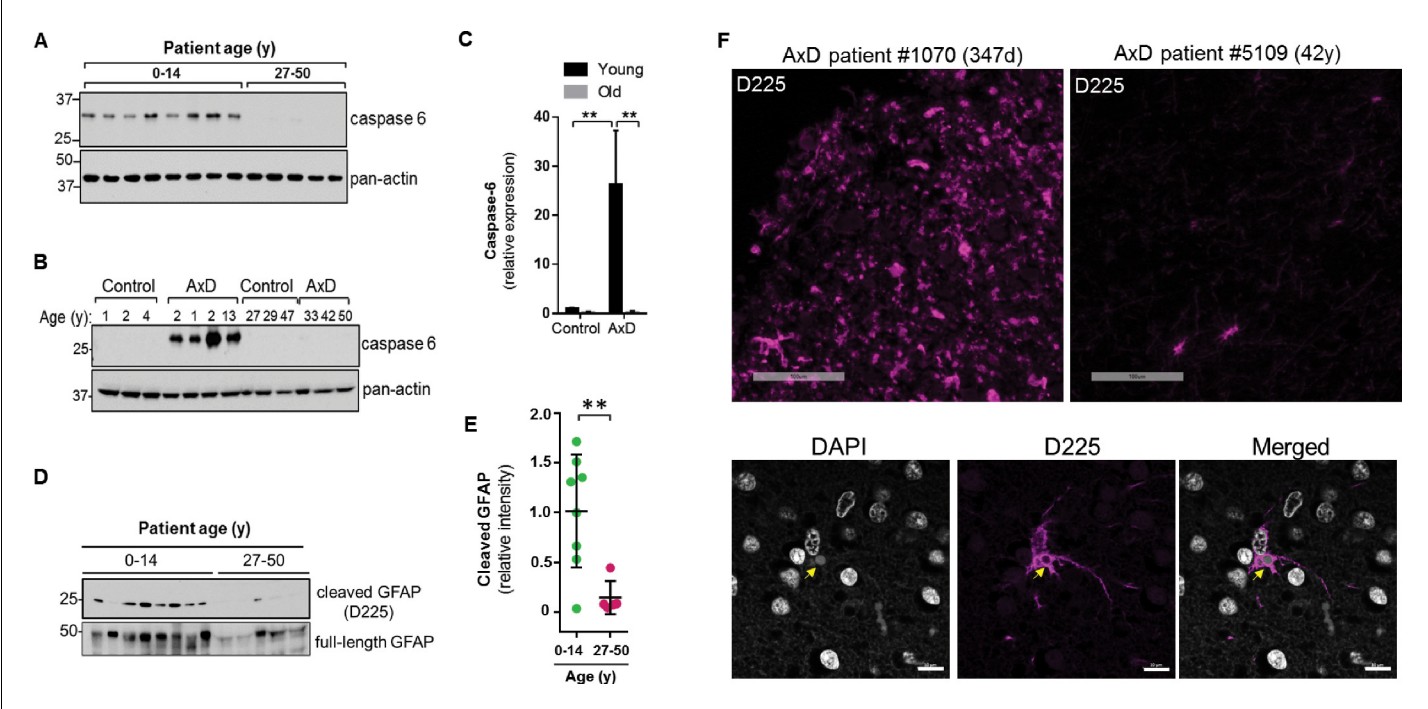

**Figure 9.** High expression of caspase-6 in young AxD patient brain tissue correlates with increased levels of cleaved GFAP. (**A**) Immunoblot for caspase-6 in total lysates from human AxD post mortem brain tissue shows that caspase-6 is upregulated in young AxD patients. Pan-actin is used as a loading control. (**B**) Immunoblot for caspase-6 in total lysates from young and old non-AxD control and AxD patient post-mortem brain tissue. Pan-actin blot serves as a loading control. (**C**) Quantification of band intensities in panel B by densitometry of caspase-6 normalized to actin. \*\*p<0.01; two-way ANOVA. (**D**) Western blotting for full-length GFAP or cleaved GFAP (D225 antibody) in HSEs from human AxD post-mortem brain tissue. (**E**) Quantification of band intensities from panel D by densitometry of D225, normalized to total GFAP (\*\*p<0.01, unpaired t-test). (**F**) Immunofluorescence staining showing widespread presence of cleaved GFAP (D225; magenta) in cerebral cortex and underlying white matter of 347 day-old child with AxD and low expression of cleaved GFAP in a 42 year old AxD patient. Wider fields of view and sections from additional patients are shown in *Figure 7—figure supplement 1*. DAPI nuclei are shown in white in bottom panels, and arrow highlights perinuclear aggregate containing cleaved GFAP and staining positively for DAPI in brain tissue from a child with AxD. Scale bar = 100 μm (top) and 10 μm (bottom).

The online version of this article includes the following figure supplement(s) for figure 9:

**Figure supplement 1.** Presence of cleaved GFAP in in post-mortem brain tissue of AxD children versus adults.

elevated in human AxD brain tissue, and subsequently validated these results using a phospho-specific antibody against the most abundant epitope (pSer13-GFAP). We demonstrate the importance of the Ser13 site for GFAP assembly in vitro and in cells. Phospho-mimetic mutation S13D completely abolished the ability of GFAP to form filaments in vitro, without leading to aggregation. In transfected SW13vim- cells, phospho mimic S13D-and S13E-GFAP mutants formed highly abnormal perinuclear aggregates that correlated with increased cleavage of GFAP by caspase-6. We detected a dramatic increase in caspase-6 expression, in association with Ser13 phosphorylation and cleavage of GFAP, in the brain tissue of AxD patients who succumbed to the disease very early in life. While the N-terminal caspase-6 fragment of GFAP promotes filament aggregation in vitro (*Chen et al., 2013*), presently we do not have direct evidence of cause and effect between caspase-6 cleavage and GFAP aggregation in AxD patient cells. Nevertheless, our current findings provide a basis for exploring PTM-based diagnostic and potential therapeutic strategies in AxD.

Our study does not address whether Ser13 phosphorylation directly promotes caspase cleavage of GFAP, or if these two PTMs are independent markers of an increased cellular stress response in AxD. One possibility is that Ser13 phosphorylation destabilizes the filament structure, thereby promoting access of caspase-6 to the rod domain Asp225 residue, where the cleavage occurs. Another

likely possibility is that the increased cleavage of GFAP is an indirect result of stress-dependent caspase-6 activation in the more severe form of AxD. This is supported by previous studies showing that AxD mutations promote activation and nuclear accumulation of p53 (*Wang et al., 2015*), which can directly induce caspase-6 expression (*MacLachlan and El-Deiry, 2002*). Future studies in AxD iPSC-astrocytes and animal models will be required to determine the timing of GFAP phosphorylation and caspase-6 activation in relationship to GFAP cleavage and aggregation.

Given our findings that pSer13-GFAP is enriched in the most aggressive form of AxD, monitoring the levels of this phospho-epitope (in addition to total GFAP) in AxD patient cerebrospinal fluid or blood may provide added sensitivity for disease activity (*Jany et al., 2015*). Phosphorylation of Ser13 by protein kinase C and cAMP-dependent protein kinase was initially described in vitro using purified recombinant GFAP (*Inagaki et al., 1990*). In the presence of active kinases, Ser-13 phosphorylation occurred in conjunction with phosphorylation at three additional sites (Thr-7, Ser-8, and Ser-34). Phosphorylation of monomeric GFAP at these sites prevented filament assembly, while phosphorylation of in vitro assembled GFAP filaments led to their disassembly (*Inagaki et al., 1990*). Using the same antibody to pSer13-GFAP that we used in this paper (clone KT13) it was later shown that Aurora-B and Rho-associated kinase phosphorylate GFAP in cultured astrocytoma cells during mitosis (*Inagaki et al., 1996*). This may bear relevance to AxD, since human and mouse AxD astrocytes with RFs display mitotic abnormalities (*Sosunov et al., 2017*). However, it was also shown using knock-in mice with the human GFAP head domain that, in vivo, the distribution of pSer13 localization was not limited to mitotic astrocytes, but that select astrocyte populations within multiple regions were pSer13 positive, such as those in the olfactory bulb, subpial regions, and subventricular zone (*Takemura et al., 2002b*; *Hagemann et al., 2005*). Interestingly, the regional distribution of pSer13 largely overlaps with areas that are known to be most enriched in RFs in the AxD mouse model (*Hagemann et al., 2005*). Therefore, this particular phosphorylation event on GFAP may occur during mitosis, or in phenotypically distinct astrocyte populations. This remains to be addressed in the future using the appropriate model systems, as over-expression studies in cancer cell lines (such as the SW13vim- cells we used here) may not be truly reflective of the signaling that occurs in astrocytes. In particular, it remains to be resolved whether phosphorylation of GFAP on Ser13 is part of a sequentially priming phosphorylation cascade involving nearby Ser16/17 (as predicted by the kinase motif analysis) or if Ser16/17 phosphorylation is unique to the SW13 over-expression system. Importantly, identifying the relevant in vivo kinase(s) that phosphorylate GFAP in human AxD may lead to potential novel interventions via kinase inhibition.

Caspase-mediated proteolysis of IF proteins is an important mechanism by which the filament networks re-organize during apoptosis. Although multiple effector caspases are capable of cleaving IF proteins, caspase-6 is frequently implicated in cleavage at a conserved motif within the linker L12 region of the rod domain, which results in the generation of two fragments of similar sizes. This was initially demonstrated to be the case for the type I keratins (*Caulín et al., 1997*), and later shown to also occur on vimentin (*Byun et al., 2001*), desmin (*Chen et al., 2003*), A-type lamins (*Ruchaud et al., 2002*), and GFAP (*Hol and Pekny, 2015*). Caspase-6 cleavage of GFAP at $_{222}$VELD$_{225}$ in vitro generates an N-terminal 26 kDa fragment and a C-terminal 24 kDa fragment. The N-terminal fragment directly impairs assembly of full-length GFAP and promotes aggregation in vitro (*Chen et al., 2013*). Using a specific antibody recognizing the N-terminal GFAP fragment (D225), we show here that GFAP cleavage is significantly increased in AxD tissues from patients presenting with an aggressive form of AxD, and that this parallels elevated expression of caspase-6. This could suggest that misregulation of caspase-6 may contribute to the severity of AxD. However, we were not able to demonstrate in cells that inhibition of caspase-6, or mutagenesis of the cleavage site on GFAP, can resolve aggregate formation. These results point to a more complex function for caspase-6, likely involving cytoskeletal remodeling in response to stress.

Indeed, caspase-6 upregulation has been reported in other neurodegenerative diseases involving protein aggregation, including Huntington's Disease (HD) and Alzheimer's Disease (AD) (*Albrecht et al., 2007*; *Graham et al., 2010*; *Guo et al., 2004*). Similar to GFAP, there is a caspase-6 cleavage site on the aggregation-prone proteins in both AD (amyloid precursor protein) and HD (huntingtin). Furthermore, in caspase-6 cleavage-resistant genetic mouse models of both HD and AD, neuronal dysfunction and degeneration are rescued (*Graham et al., 2006*; *Galvan et al., 2006*; *Saganich et al., 2006*). Caspase-6 can promote neurodegeneration via induction of neuronal apoptosis or axon pruning (*Geden et al., 2019*). However, the functions of caspase-6 in astrocytes are

not clear. In the context of human AxD it still remains to be determined which astrocyte populations express caspase-6, and whether it promotes apoptosis or performs a non-apoptotic role, such as sculpting the cytoskeletal architecture in reactive astrocytes. Based on our demonstration that caspase-6 localizes within the perinuclear GFAP inclusions in the AxD iPSC-astrocytes, it is intriguing to speculate that, similar to keratin inclusions in epithelial cells (*Dinsdale et al., 2004*), RFs sequester active caspases away from other cellular substrates and may protect reactive astrocytes from apoptosis.

Recently, iPSC-derived patient astrocyte models have emerged as an important system for dissecting the cellular mechanisms in AxD. For example, these novel tools have revealed that AxD astrocytes have defects in the secretory pathway, impaired ATP release, and attenuated calcium waves (*Jones et al., 2018*); that they inhibit oligodendrocyte precursor cell proliferation (*Li et al., 2018*), providing a potential mechanistic explanation for the degeneration of white matter observed in patients; and that they have defects in mechanotransduction signaling pathways (*Wang et al., 2018*). A novel aspect of the AxD astrocyte cell model that we generated in our study is the perinuclear accumulation of pSer13-GFAP that was associated with prominent nuclear abnormalities. As such, these patient-derived cells replicate a key phenotypic characteristic of RF-bearing AxD astrocytes in vivo, since nuclear invaginations have been described in electron microscopy studies of AxD mouse models and AxD patient cortex (*Sosunov et al., 2017*). Another important parallel is that the GFAP inclusions we observe in the AxD patient astrocytes in vitro stain positive for DAPI, and it was shown that DAPI is a reliable and sensitive marker of RFs in human and mouse brain (*Sosunov et al., 2017*). Therefore patient-derived iPSC-astrocytes provide a unique model system to investigate cytoplasmic-nuclear mechanics in AxD.

Invaginations of the nucleus, such as those we observe here, have been described in physiological and pathological states (*Malhas et al., 2011*). Control of nuclear shape is critical for regulation of gene expression and response to mechanotransduction signals (*Uhler and Shivashankar, 2017*). The effects of impaired nuclear morphology can be very severe, as evidenced by mutations in lamin A that lead to defective nuclear morphology in Hutchinson-Gilford Progeria Syndrome (HGPS), where patients experience accelerated aging (*Eriksson et al., 2003*). An elegant study combining multiple 3D imaging strategies established a direct link between intermediate filaments, actin and the nuclear envelope within nuclear invaginations, and genetic evidence indicates that filamentous actin may play a role in generating these structures (*Jorgens et al., 2017*; *Frost et al., 2016*). It is hypothesized that nuclear invaginations provide localized control of gene expression and nuclear-cytoplasmic transport deep within the nucleus, since they have been found to contain calcium receptors and nuclear pores (*Malhas et al., 2011*). Our study provides the first link between abnormal cytoplasmic PTM processing and perinuclear accumulation of mutant GFAP with nuclear defects, setting the stage to address how nucleo-cytoskeletal coupling is adversely impacted by defective IF proteostasis in AxD and related human diseases.

# Materials and methods

**Key resources table**

| Reagent type (species) or resource | Designation | Source or reference | Identifiers | Additional information |
|---|---|---|---|---|
| Gene (human) | *GFAP* | NA | Gene ID: 2670 | |
| Cell line (human) | R239C-GFAP fibroblasts from AxD patient | Coriell Institute | GM16825 | *Brenner et al. (2001)* |
| Cell line (human) | R239C-GFAP induced pluripotent stem cells | Generated in the study | | Generation of the AxD iPSCs (R239C-GFAP) is described in the Materials and methods below. Cells can be obtained by contacting the corresponding author. |

*Continued on next page*

*Continued*

| Reagent type (species) or resource | Designation | Source or reference | Identifiers | Additional information |
|---|---|---|---|---|
| Cell line (human) | R239R-GFAP isogenic control induced pluripotent stem cells | Generated in the study | | Generation of the AxD iPSCs (R239C-GFAP) is described in the Materials and methods below. Cells can be obtained by contacting the corresponding author. |
| Cell line (human) | SW13 vim- | Sarria, A.J et al. J Cell Sci 1994 | | |
| Biological sample (human) | Human brain specimens | NIH NeuroBioBank | Listed in *Supplementary files 1* and *2* | |
| Antibody | rabbit anti-GFAP | Agilent/DAKO | Clone Z0334 | Dilution = 1:10,000 immunoblot, 1:500 immunofluorescence |
| Antibody | rabbit anti-caspase-6 | Cell Signaling Technology | Cat# 9762 | Dilution = 1:1000 immunoblot |
| Antibody | rabbit anti-caspase-6 | abcam | Cat# ab185645 | Dilution = 1:100 immunofluorescence |
| Antibody | rabbit anti-D225 | PMID: 24102621 | | gift from Dr Ming Der Perng, Dilution = 1:5000 immunoblot (overnight), 1:150 immunofluorescence |
| Antibody | mouse anti-GFAP | Sigma | Clone GA5, Cat# G3893 | Dilution = 1:3000 immunoblot, 1:300 immunofluorescence |
| Antibody | mouse anti-pSer13 GFAP | PMID: 8647894 | | gift from Dr Masaki Inagaki, Dilution = 1:500 immunoblot (overnight), 1:20 immunofluorescence |
| Antibody | mouse anti-pan Actin | NeoMarkers | Cat# MS-1295 | Dilution = 1:3000 immunoblot |
| Antibody | mouse anti-Tra-1–60 | ThermoFisher | Cat# 41–1000 | Dilution = 1:300 |
| Antibody | mouse anti-Tra-1–81 | ThermoFisher | Cat# 41–1100 | Dilution = 1:300 |
| Antibody | mouse anti-SSEA4 | ThermoFisher | Cat#41–4000 | Dilution = 1:300 |
| Antibody | rabbit anti-Oct4 | abcam | Cat# ab19857 | Dilution = 1:40 |
| Antibody | rabbit anti-Sox2 | ThermoFisher | Cat# 48–1400 | Dilution = 1:125 |
| Antibody | Alexa 488-conjugated goat anti-mouse | ThermoFisher | Cat# A32723 | Dilution = 1:500 |
| Antibody | Alexa 488-conjugated goat anti-rabbit | ThermoFisher | Cat# A32731 | Dilution = 1:500 |
| Antibody | Alexa 594-conjugated goat anti-mouse | ThermoFisher | Cat# A32742 | Dilution = 1:500 |
| Antibody | Alexa 594-conjugated goat anti-rabbit | ThermoFisher | Cat# A32740 | Dilution = 1:500 |
| Recombinant DNA reagent | pCMV6-XL6-GFAP | Origene | Cat# SC118873 | |
| Peptide, recombinant protein | TrueCut Cas9 Protein v2 | ThermoFisher | Cat# A36499 | |
| Commercial assay or kit | Precision gRNA Synthesis Kit | ThermoFisher | Cat# A29377 | |

*Continued on next page*

*Continued*

| Reagent type (species) or resource | Designation | Source or reference | Identifiers | Additional information |
|---|---|---|---|---|
| Commercial assay or kit | Agilent Quikchange II | Agilent | Cat# 200524 | |
| Commercial assay or kit | Rneasy Kit | Qiagen | Cat# 74104 | |
| Commercial assay or kit | Taqman Scorecard | ThermoFisher | Cat# A15870 | |
| Chemical compound, drug | ECL Reagents | Perkin Elmer | NEL103E001EA | |
| Chemical compound, drug | Ac-VEID-CHO | Millipore Sigma | A6339 | |
| Software, algorithm | CRISPR off-target | PMID: 27380939 | http://crispor.tefor.net/ | |

## Antibodies

The following antibodies were used: rabbit anti-GFAP (DAKO Agilent, Santa Clara, CA, Z0334), rabbit anti-caspase-6 (Cell Signaling Technologies, Danvers, MA, 9762), rabbit anti-Caspase-6 (abcam, Cambridge, UK, ab185645), rabbit anti-D225 (*Chen et al., 2013*), mouse anti-GFAP (Sigma, GA5), mouse anti-pSer13-GFAP (KT13 [*Sekimata et al., 1996*]), mouse anti-pan Actin, mouse anti-Tra-1–60, mouse anti-SSEA4, rabbit anti-Oct4, rabbit anti-Sox2, and Alexa 488- and Alexa 594-congujated goat anti mouse or rabbit antibodies (Thermo Fisher Scientific, Waltham, MA).

## Cell lines

SW13vim- cells were provided by Dr Bishr Omary and cultured in DMEM with 10% fetal bovine serum and 1% penicillin-streptomycin. Authentication of the cell line was done by short tandem repeat (STR) profiling by ATCC. Fibroblasts from a male 6 year old type I AxD patient were obtained from the Coriell institute (Camden, NJ). Sanger sequencing was performed to confirm the AxD mutation was present in the cells (c.715C > T; p.Arg239Cys). The cell lines used tested negative for mycoplasma contamination, as assayed using the Universal Mycoplasma Detection Kit (ATCC 30–1012K).

## Human brain tissues

De-identified post-mortem fresh-frozen and fixed AxD patient and control brain tissues were provided by the NIH NeuroBioBank and are described in *Supplementary files 1* and *2*.

## Mass spectrometry

*Sample Preparation:* HSEs from AxD patient post-mortem brain cortex tissue were prepared as described previously (*Battaglia et al., 2017*; *Snider and Omary, 2016*) and in *Figure 1—figure supplement 1*, then subjected to SDS-PAGE followed by Coomassie stain. Bands corresponding to GFAP were excised and the proteins were reduced, alkylated, and in-gel digested with trypsin overnight at 37 ˚C. Peptides were extracted, desalted with C18 spin columns (Pierce – Thermo Fisher Scientific) and dried via vacuum centrifugation. Peptide samples were stored at −80˚C until further analysis. *LC-MS/MS Analysis:* The peptide samples were analyzed by LC/MS/MS using an Easy nLC 1200 coupled to a QExactive HF mass spectrometer (Thermo Fisher Scientific). Samples were injected onto an Easy Spray PepMap C18 column (75 μm id ×25 cm, 2 μm particle size) (Thermo Fisher Scientific) and separated over a 1 hr method. The gradient for separation consisted of 5–40% mobile phase B at a 250 nl/min flow rate, where mobile phase A was 0.1% formic acid in water and mobile phase B consisted of 0.1% formic acid in 80% ACN. The QExactive HF was operated in data-dependent mode where the 15 most intense precursors were selected for subsequent fragmentation. Resolution for the precursor scan (m/z 300–1600) was set to 120,000 with a target value of $3 \times 10^6$ ions. MS/MS scans resolution was set to 15,000 with a target value of $1 \times 10^5$ ions. The normalized collision energy was set to 27% for HCD. Dynamic exclusion was set to 30 s, peptide match was set to preferred, and precursors with unknown charge or a charge state of 1 and $\geq 7$ were

excluded. *Data Analysis:* Raw data files were processed using Proteome Discoverer version 2.1 (Thermo Fisher Scientific). Peak lists were searched against a reviewed Uniprot human database, appended with a common contaminants database, using Sequest. The following parameters were used to identify tryptic peptides for protein identification: 10 ppm precursor ion mass tolerance; 0.02 Da product ion mass tolerance; up to two missed trypsin cleavage sites; phosphorylation of Ser, Thr and Tyr were set as variable modifications. The ptmRS node was used to localize the sites of phosphorylation. Peptide false discovery rates (FDR) were calculated by the Percolator node using a decoy database search and data were filtered using a 5% FDR cutoff. The peak areas for the identified peptides were extracted and used for relative quantitation across samples.

## Site directed mutagenesis, in vitro assembly, transfections, and immunofluorescence

Mutagenesis of GFAP (Origene, Rockville, MD, in vector CMV6-XL6) was performed using the Quik-Change II mutagenesis kit (Agilent) to generate the designated point mutants. Sanger sequencing of the entire coding sequence of GFAP was performed to confirm the wild-type and mutant sequences. We used established procedures for the purification and in vitro assembly of GFAP (*Perng et al., 2016*). For transfections, lipofectamine 2000 was used according to the supplier instructions (Invitrogen, Thermo Fisher Scientific, Carlsbad, CA), and experiments were performed 20–24 hr after transfection. For immunofluorescence, cells were fixed in methanol at −20°C for 10 min, washed three times in PBS and incubated in blocking solution (2.5% bovine serum albumin, 2% normal goat serum in PBS) for 1 hr at room temperature. Primary antibodies were diluted into blocking buffer and incubated overnight at 4°C. The next day, cells were washed 3 times in PBS and incubated with Alexa Fluor-conjugated secondary antibodies diluted into blocking buffer for 1 hr at room temperature. Cells were washed 3 times in PBS, incubated in DAPI for 5 min, washed 3 times and mounted in Fluoromount-G (SouthernBiotech, Birmingham, AL) overnight. Cells were imaged on Zeiss 880 confocal laser scanning microscope using a 63x (1.4 NA) oil immersion objective (Zeiss, Jena, Germany).

## Preparation of protein lysates and western blotting

High salt extracts (HSEs) and triton-X (TX) lysates were prepared as previously described (*Battaglia et al., 2017*). Total lysates were prepared by homogenizing 25 mg tissue directly into hot 2X Tris-Glycine SDS Sample Buffer (Thermo Fisher Scientific) and heating for 5 min at 95°C. Immunoblotting was performed as previously described (*Trogden et al., 2018*). Briefly, samples were resolved on 4–20% gradient SDS-PAGE gels transferred onto activated polyvinylidene difluoride membranes at 40V overnight. The transferred gels were routinely stained with Coomassie blue and the membranes were blocked in 5% non-fat milk in 0.1% tween 20/PBS (PBST). Post-transfer Coomassie-stained gels served as another loading control where the levels of housekeeping protein (actin) varied (*Figure 4E*). For immunoblotting, the membranes were incubated with the appropriate primary antibody diluted in 5% milk/PBST, with the exception of KT13, which was incubated in 5% bovine serum albumin/PBST for blocking, primary antibodies and secondary antibodies. Antibodies were detected using ECL reagents (PerkinElmer Life Sciences, Hopkinton, MA). For 2D gel analysis, HSEs were dissolved in 2-D starter kit rehydration/sample buffer (Biorad; 1632106) for separation by isoelectric focusing (IEF). Immobilized pH gradient (IPG) strips (Biorad; 11 cm; pH 4–7; 1632015) were passively rehydrated in 2-D starter kit rehydration/sample buffer overnight. Cup loading method was employed to load the protein samples in cathode side (as isoelectric point of GFAP is 5.2) of the Protean IEF cell tray (Biorad; 1654020). The IEF separation was done using 72000 vh. After IEF separation the protein samples were further separated based on molecular weight using SDS-PAGE gel by applying constant 90 volts.

## Cellular reprogramming, characterization and karyotyping of iPSCs

Skin fibroblasts were reprogrammed under feeder free conditions using Cytotune –iPS 2.0 Sendai Reprogramming kit and individual iPSC clones were picked for propagation in culture for 10 passages. To confirm stemness and differentiation capabilities of reprogrammed and edited iPSCs, we used the qPCR based TaqMan human Pluripotent Stem Cell Scorecard Panel (Thermo Fisher Scientific). iPSCs were differentiated into all three germ layers using STEMdiff Trilineage Differentiation Kit (StemCell Technologies, Vancouver, Canada), and a monolayer-based protocol was used to

directly differentiate hES cells in parallel into the three germ layers (~1 week). Non-differentiated and differentiated cells were lysed and total RNA purified using the RNeasy kit (QIAGEN). RNA reverse transcription was performed following the Taqman Scorecard's manufacture guidelines and the qRT-PCR was carried out using the QuantStudio 7 Flex Real-Time PCR system. The TaqMan PCR assay combines DNA methylation mapping, gene expression profiling, and transcript counting of lineage marker genes (*Bock et al., 2011*). Reprogrammed and edited iPSCs were submitted to a standard G-band analysis consisting of 20 metaphase spreads. The analysis (carried out by Karyologic Inc) can identify gender, chromosome number, and detect aberrations that include trisomies, monosomies, deletions, insertions, translocations, duplications, breaks, polyploidy, among others. No abnormalities were found in our cell lines (*Figure 4—figure supplement 2A*).

## CRISPR/Cas9 genome editing

We used the TrueCut Cas9 Protein V2, sgRNAs and the Neon Transfection system (Thermo Fisher Scientific) to edit iPSCs. The recombinant TrueCut Cas9 V2 was diluted in resuspension buffer R provided in the kit and mixed with 900 ng of sgRNA and 2700 ng of single-stranded donor oligonucleotide, incubated 15 min at room temperature and then a total of $3 \times 10^5$ iPSCs were electroporated with the ribonucleoprotein mix. Seventy-two hours after electroporation, cells were dissociated into single cells, diluted, and seeded on Matrigel-coated 96-well plates. Single-cell colonies were selected after two weeks and tested for gene correction. Genomic DNA of single clones was extracted and the gene of interest amplified by PCR using allele specific primers. Sanger sequencing of positive clones demonstrated single or double allele gene correction. Off-target sites within the exons of genes were predicted via selection of the top candidates using the MIT software (CRISPR. mit.edu). The analysis was performed via PCR of 400 bp fragments, which flanked the predicted off-target cut site followed by Sanger sequencing. The chromatograms for edited clones were compared to sequences from the original AxD patient cells.

## iPSC culture and astrocyte differentiation

iPSCs were maintained on Matrigel in StemFlex medium (Thermo Fisher Scientific) and passaged every 3–4 days with 0.5 mM EDTA dissociation solution. iPSCs were differentiated into neural progenitor cells (NPC) using an embryoid body (EB) protocol. Briefly, iPSCs at 80% confluence were collected, resuspended in Neural Induction Medium (NIM, StemCell Technologies) and seeded on one well of an Aggrewell 800 plate (StemCell Technologies) at $3 \times 10^6$ cells per well. At day five, EBs were seeded on poly-ornithine and laminin (PLO/LAM)-coated dishes in NIM. Rosette selection was performed after 12 days using Rosette Selection Reagent (StemCell Technologies). NPCs were expanded for 7 days in Neural Progenitor Medium (StemCell Technologies). NPCs were then differentiated into astrocyte precursors by seeding dissociated single cells at $1 \times 10^5$ cells/cm$^2$ density on PLO/LAM dishes in STEMdiff astrocyte differentiation medium (StemCell Technologies). Astrocyte precursors were maintained for 20 days with medium changes every 48 hr and splitting every week with Accutase (Millipore, Burlington, MA). Astrocytes were expanded for up to 120 days in STEMdiff astrocyte maturation medium (StemCell Technologies).

## Transmission electron microscopy

AxD iPSC-astrocytes grown on a polystyrene dish were fixed in 2.5% glutaraldehyde in 0.1M sodium cacodylate buffer, pH 7.4, for one hour at room temperature and stored at 4°C. The cells were washed 3 times in 0.1M sodium cacodylate buffer followed by post-fixation in 1% buffered osmium tetroxide for 1 hr. After three washes in deionized water, the cells were dehydrated in ethanol, infiltrated and embedded in situ in PolyBed 812 epoxy resin (Polysciences, Inc, Warrington, PA). The cell monolayer was sectioned *en face* to the substrate with a diamond knife and Leica UCT Ultramicrotome (Leica Microsystems, Inc, Buffalo Grove, IL). Ultrathin sections (70 nm) were mounted on 200 mesh copper grids and stained with 4% uranyl acetate and lead citrate. The sections were observed and digital images were taken using a JEOL JEM-1230 transmission electron microscope operating at 80kV (JEOL USA, Inc, Peabody, MA) equipped with a Gatan Orius SC1000 CCD Digital Camera (Gatan, Inc, Pleasanton, CA).

## Acknowledgements

This work was supported by grants from Elise's Corner Fund and the United Leukodystrophy Foundation (to NTS), the National Science Foundation Graduate Research Fellowship (to RAB) and the Department of Cell Biology and Physiology at UNC-Chapel Hill. The authors thank Kristen White, Microscopy Services Laboratory, for her assistance with EM specimen preparation. The Microscopy Services Laboratory, Department of Pathology and Laboratory Medicine, is supported in part by P30 CA016086 Cancer Center Core Support Grant to the UNC Lineberger Comprehensive Cancer Center. The authors thank all the patients and their families for donating tissue that enabled this research.

## Additional information

### Competing interests

Namritha Ravinder, Erik Willems, Rhonda A Newman: are paid employees of ThermoFisher Scientific, whose products were used to complete parts of the study. ThermoFisher Scientific had no role in the study design, data analysis, decision to publish, or preparation of the manuscript. Natasha T Snider: is a member of the Scientific Advisory Board for Elise's Corner Fund, which supported part of this work. The other authors declare that no competing interests exist.

### Funding

| Funder | Grant reference number | Author |
| --- | --- | --- |
| Elise's Corner Fund | Research Grant | Natasha T Snider |
| United Leukodystrophy Foundation | Research Grant | Natasha T Snider |
| National Science Foundation | Graduate Research Fellowship | Rachel A Battaglia |
| University of North Carolina at Chapel Hill | Department of Cell Biology and Physiology | Natasha T Snider |
| National Institutes of Health | P30 CA016086 | Victoria J Madden |

The funders had no role in study design, data collection and interpretation, or the decision to submit the work for publication.

### Author contributions

Rachel A Battaglia, Conceptualization, Data curation, Formal analysis, Funding acquisition, Validation, Investigation, Methodology, Writing—original draft, Writing—review and editing; Adriana S Beltran, Formal analysis, Supervision, Investigation, Methodology, Writing—review and editing; Samed Delic, Formal analysis, Validation, Investigation, Methodology, Writing—review and editing; Raluca Dumitru, Data curation, Formal analysis, Validation, Methodology, Writing—review and editing; Jasmine A Robinson, Formal analysis, Validation, Investigation, Writing—review and editing; Parijat Kabiraj, Data curation, Formal analysis, Investigation, Methodology, Writing—review and editing; Laura E Herring, Data curation, Formal analysis, Methodology, Writing—review and editing; Victoria J Madden, Erik Willems, Formal analysis, Investigation, Methodology, Writing—review and editing; Namritha Ravinder, Rhonda A Newman, Methodology, Writing—review and editing; Roy A Quinlan, Resources, Supervision, Investigation, Methodology, Writing—review and editing; James E Goldman, Resources, Formal analysis, Methodology, Writing—review and editing; Ming-Der Perng, Masaki Inagaki, Resources, Writing—review and editing; Natasha T Snider, Conceptualization, Resources, Data curation, Formal analysis, Supervision, Funding acquisition, Validation, Investigation, Visualization, Methodology, Writing—original draft, Project administration, Writing—review and editing

## Author ORCIDs

Victoria J Madden  http://orcid.org/0000-0002-7909-7743
Roy A Quinlan  https://orcid.org/0000-0003-0644-4123
Natasha T Snider  https://orcid.org/0000-0001-7663-4585

## Decision letter and Author response

Decision letter https://doi.org/10.7554/eLife.47789.sa1
Author response https://doi.org/10.7554/eLife.47789.sa2

## Additional files

### Supplementary files

- Supplementary file 1. Donor information for AxD post-mortem human brain specimens.
- Supplementary file 2. Donor information for control (non-AxD) post-mortem human brain specimens.
- Supplementary file 3. Summary from off-target sequencing from CRISPR/Cas9 editing.
- Supplementary file 4. GFAP phosphorylation motifs and candidate kinases.
- Transparent reporting form

### Data availability

All data generated or analyzed during this study are included in the manuscript and supporting files. Source data files for mass spectrometry results in Figure 2 and Figure 7 are provided in Figure 2—source data 1 and Figure 7—source data 1, respectively.

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
