## [Decision Letter]

**Acceptance summary:**

We are particularly excited about the findings that Alexander Disease-causing mutations perturb key post-translational modifications (PTMs) on GFAP. The selective phosphorylation of GFAP-Ser13 in patients who died young raises the hypothesis that Ser13 phosphorylation and increased GFAP proteolysis by caspase-6 facilitated GFAP aggregation, which leads to disease state.

**Decision letter after peer review:**

Thank you for submitting your article "Phosphorylation-dependent proteolysis perturbs GFAP proteostasis, implicating caspase-6 in early onset Alexander Disease" for consideration by *eLife*. Your article has been reviewed by three peer reviewers, and the evaluation has been overseen by a Reviewing Editor and Huda Zoghbi as the Senior Editor. The following individuals involved in review of your submission have agreed to reveal their identity: W Christopher Risher (Reviewer #2).

The reviewers have discussed the reviews with one another and the Reviewing Editor has drafted this decision to help you prepare a revised submission.

Summary:

Battaglia and co-authors have identified a posttranslational modification of the GFAP protein and potential mechanism involving caspase-6 that appear to correlate with disease severity in the rare but lethal disorder Alexander disease (AxD). In this report post-mortem samples from individuals with known GFAP mutations were analyzed for GFAP-PTM by mass spectrometry. Phospho-Ser13 was identified primarily in samples from AxD patients with early onset, regardless of the mutational variant. Subsequent transfection experiments with Ser13 phospho- and de-phospho mimics further suggest that Ser13 phosphorylation promotes protein aggregation. In addition, the authors show that pSer13-GFAP along with caspase-6 localize to aggregates found in astrocytes derived from AxD iPSCs, and that the phospho-mimic S13D leads to caspase-6 cleavage of GFAP, similar to AxD mutations. Finally, they show that pSer13, caspase-6 protein, and caspase-6 cleaved GFAP are present in brains from early onset AxD patients.

Essential revisions:

All three reviewers found the question interesting and the results potentially significant.

1) The major critique from all three reviewers is that the causality between phosphorylation of GFAP and cleavage of caspase-6 in relation to the pathology of AxD are not explored. For example, does blocking phosphorylation of this site on GFAP in the patient iPSC astrocytes prevent/reverse the GFAP aggregation and perinuclear accumulation of GFAP? Are there known effects on astrocyte health/reactivity in addition to GFAP aggregation that can be monitored, to see if they are also rescued? Experiments along these lines would greatly strengthen the study.

Specific comments along this line include "The key pieces of evidence suggesting cause and effect are the transfection experiments with phopho-mimetic constructs. Although N-terminal substitutions are rare in AxD, how do we know that the S13D mutation does not cause GFAP aggregation in itself (multiple AxD associated mutations have been identified at Ser385, 393 and 398)? Did the authors try other substitutions, e.g. S13V, S13E (or D225E to prevent caspase-6 cleavage)? Since it is known that over-expression of wild-type GFAP causes aggregation, how are GFAP expression levels controlled between transfections? If phosphorylation of GFAP-Ser13 promotes caspase-6 cleavage and GFAP aggregation, shouldn't elimination of pSer13 alleviate aggregate formation in the R79H/S13A transfection? Would caspase-6 inhibitors (Ac-VEID-CHO) prevent aggregation in either the SW13vim-cells or iPSC astrocytes? The authors claim in the Discussion that impairment of GFAP assembly by the S13D phospho-mimic is due to caspase-6 cleavage, but this link is not demonstrated by the data shown. "

2) The Western blots in the figures (Figure 3E, Figure 6B, Figure 7B) used Coomassie staining as the loading control, but no details were given. Given that antibody labeling of Coomassie-stained gels is notoriously difficult, did the authors use different gels for determining protein amounts vs. blotting? This would not be an appropriate normalization method for the actual blotted membranes, and would therefore need to be replaced. If these Coomassie controls were done on the same gels that went on to be blotted for antibodies, please provide technical details in the Materials and methods, as this is a non-standard approach.

[Editors' note: further revisions were requested prior to acceptance, as described below.]

Thank you for resubmitting your work entitled "Phosphorylation-dependent proteolysis perturbs GFAP proteostasis, implicating caspase-6 in early onset Alexander Disease" for further consideration at *eLife*. Your revised article has been favorably evaluated by Huda Zoghbi (Senior Editor), a Reviewing Editor, and three reviewers.

The reviewers agreed that the manuscript has been improved and is a strong candidate for *eLife*, but there are some remaining issues that need to be addressed before acceptance, as outlined below:

The main remaining issue is on how strong the conclusions can be drawn. Please make modifications to the text and potentially to the title to reflect the concerns of reviewer 3.

Reviewer #1:

The authors have added a number of new experiments to the paper that strengthen the conclusion linking phosphorylation of GFAP to caspase-6 cleavage of GFAP and to the pathology of Alexander disease. While major point 1 is not fully addressed (determining whether these changes are a cause or a consequence of more severe disease mutations), the paper does make significant new findings in the area of Alexander disease that warrant publication in *eLife*.

Reviewer #2:

The authors have satisfactorily addressed all of my major and minor concerns; I have no further issues. Additionally, the mechanistic insight provided by the new experiments and analyses has significantly elevated the impact of the study.

Reviewer #3:

The authors have made a significant effort in adding important data to address the reviewers' concerns and questions. These include in vitro assembly analysis for the GFAP S13D phospho-mimic, the addition of the S13E phospho-mimic and analysis of GFAP-R79H phosphorylation sites in transfection experiments (SW13vim-), and both genetic and pharmacological inhibition of caspase-6 to assess the effects of GFAP cleavage on protein aggregation in SW13vim- cells. Important controls were also added for the iPS cell and AxD patient experiments. Most of these additions have strengthened the manuscript and support a connection between GFAP phosphorylation and aggregation. However, the new data showing caspase-6 inhibition in itself disrupts GFAP filament organization and fails to rescue S13D aggregation complicate interpretation of the results, and evidence for direct cause and effect between caspase-6 GFAP cleavage and protein aggregation is still lacking. The findings in this report are important in identifying PTMs as new markers of disease severity and should be published, but as noted in the original review, the authors should be more conservative with their conclusions.

The authors indicate a direct link between GFAP-pSer13, caspase-6 cleavage, and protein aggregation in the title, Introduction and Discussion. A major question is still whether the observed phenomena contribute to disease severity or are downstream in the stress response and simply reflect the more severe pathology in early onset AxD. The following 3 areas contribute to this concern:

1) In agreement with a more severe phenotype and astrocyte response, GFAP levels are higher in early onset AxD cases compared to those with late onset (Figure 9D). GFAP has been shown to be phosphorylated in reactive gliosis associated with certain injury models (Takemura et al., 2002), and the overall damage response could have a broader effect in activating caspase-6; p53 for example is activated in AxD (Wang et al., 2015) and is known to transactivate caspase-6 (MacLachlan and El-Deiry, 2002). The authors hypothesize that GFAP phosphorylation directly affects caspase-6 accessibility, but aggregation of GFAP-S13D and S13E in SW13vim- cells likely causes a cellular stress response that could also indirectly activate caspase-6. While I agree with the authors that investigating these mechanisms is beyond the scope of the current manuscript, they should address this possibility in their discussion and conclusions.

2) New experiments to inhibit caspase-6 cleavage of GFAP produce ambiguous results. Inhibition of caspase-6 alone leads to GFAP bundling in SW13vim- cells and in combination with the S13D phospho-mimic, aggregates are not resolved. Western analysis shows clearance of a 100 kD (hmm) band, but a ~200 kD band remains and no western data is shown to demonstrate the level of WT-GFAP oligomerization induced by caspase-6 inhibition. As the authors suggest in the discussion, the role of caspase-6 activation is complex and may also be involved in cytoskeletal reorganization.

3) The supplemental data added to Figure 7 introduce important evidence for a link between GFAP phosphorylation, aggregation and caspase-6 cleavage. Immunoblotting shows de-phospho-mimics of serine residues (S to A) within the S13-S17 motif of mutant R79H GFAP eliminate high molecular weight oligomerization and significantly reduce caspase-6 cleavage fragments. However, these data are complicated by the limited contribution of pSer13 in GFAP-R79H aggregation in SW13vim- cells and the more dominant effects of pS16 and pS17, which have not been otherwise explored in this report. Although the SW13vim- cell line is commonly used to test the effects of GFAP mutation, these results demonstrate the limitations of the system for these experiments. Future work should include similar experiments in the iPSC derived astrocytes.

---

## [Author Response]

Essential revisions:All three reviewers found the question interesting and the results potentially significant.1) The major critique from all three reviewers is that the causality between phosphorylation of GFAP and cleavage of caspase-6 in relation to the pathology of AxD are not explored. For example, does blocking phosphorylation of this site on GFAP in the patient iPSC astrocytes prevent/reverse the GFAP aggregation and perinuclear accumulation of GFAP? Are there known effects on astrocyte health/reactivity in addition to GFAP aggregation that can be monitored, to see if they are also rescued? Experiments along these lines would greatly strengthen the study.

We present new experimental evidence (Figure 7—figure supplement 1; Figure 7—source data 1) and kinase motif-based predictions (Supplementary file 4) revealing the complexity of this PTM mechanism on GFAP (e.g. additional effects of nearby phosphorylation sites S16/17). Although we are actively working to identify the critical kinase(s) involved in the phosphorylation of S13 (through candidate and unbiased approaches), we currently do not have strong and conclusive evidence for the specific involvement of a major kinase that we can selectively inhibit to measure the effects. However, we fully agree with the reviewers that this is of key importance and it is an avenue that we are actively exploring, using a combination of in vitro and in vivo approaches to find the most promising target for intervention.

Specific comments along this line include "The key pieces of evidence suggesting cause and effect are the transfection experiments with phopho-mimetic constructs. Although N-terminal substitutions are rare in AxD, how do we know that the S13D mutation does not cause GFAP aggregation in itself (multiple AxD associated mutations have been identified at Ser385, 393 and 398)?

To address this question, we purified WT-, S13A-, and S13D-GFAP and performedin vitro filament assembly assays. As shown in new Figure 3C, the S13D mutant is unable to assemble into filaments, and it does not aggregate on its own. This is in contrast to the phenotype of the N-terminal caspase-cleaved fragment, which does aggregate on its own (please see Figure 2B of Chen et al., 2013).

Did the authors try other substitutions, e.g. S13V, S13E (or D225E to prevent caspase-6 cleavage)?

We now include new data on the effects of an S13E substitution, which phenocopied S13D with respect to aggregation (Figure 3A-B) and caspase cleavage (Figure 7A-B). We also generated D225E mutant alone, and in combination with S13D. We present new data to show that D225E mutation essentially eliminates S13D-GFAP cleavage, and that this correlates with a decrease in aggregation (Figure 8A-D). We note that D225E on its own interfered with GFAP filament formation, which is not surprising as caspase-cleavage is an important mechanism used to re-organize IFs.

Since it is known that over-expression of wild-type GFAP causes aggregation, how are GFAP expression levels controlled between transfections?

For all transfection experiments, we initially carried out titration of WT-GFAP cDNA amounts (250ng-1µg) in a 24-well format using three different transfection reagents (Lipofectamine 2000, 3000, and LTX). We combined this with a time-course experiments (24, 48, 72h). Based on these initial optimization experiments, we selected to use Lipofectamine 2000/500ng cDNA/24hr conditions. Under these conditions, there is minimal aggregation of WT-GFAP (based on protein solubility in TritonX-100 buffer and filament organization). In contrast, under these same conditions, AxD-associated GFAP mutants are significantly less soluble and form the expected aggregates. We now show this as a supplement (Figure 3—figure supplement 1).

If phosphorylation of GFAP-Ser13 promotes caspase-6 cleavage and GFAP aggregation, shouldn't elimination of pSer13 alleviate aggregate formation in the R79H/S13A transfection?

We present new data that R79H in the over-expression system is highly phosphorylated at nearby serines 16/17 (Figure 7—figure supplement 1; Figure 7—source data 1). Mutagenesis of these residues to non-phosphorylatable alanines decreases GFAP cleavage and oligomerization of the R79H mutant. Therefore, our current hypothesis is that phosphorylation of S13 is a coordinated event that works in tandem with nearby phosphorylation sites. This is also predicted by the kinase motif analysis (Supplementary file 4). Another contributing factor may be that the S13A substitution leads to the formation of abnormal filaments, based on the in vitro assembly assay (Figure 3C). Given these results, it is not surprising that S13A does not alleviate R79H aggregation.

Would caspase-6 inhibitors (Ac-VEID-CHO) prevent aggregation in either the SW13vim-cells or iPSC astrocytes?

These data are shown in new Figure 8E-G. In the presence of the caspase-6 inhibitor, S13D aggregates in the SW13vim- cells were reduced in size, but we observed more filament bundles (consistent with previous studies on other IFs showing that caspase cleavage is required for filament re-organization). Additionally, the presence of a high-molecular-mass (100kDa) S13D-GFAP oligomer was significantly reduced by caspase-6 inhibition.

The authors claim in the Discussion that impairment of GFAP assembly by the S13D phospho-mimic is due to caspase-6 cleavage, but this link is not demonstrated by the data shown.

We modified this section of the Discussion to include the new results with the recombinant protein and propose a potential mechanism regarding the aggregation mechanism of the S13D mutant.

2) The Western blots in the figures (Figure 3E, Figure 6B, Figure 7B) used Coomassie staining as the loading control, but no details were given. Given that antibody labeling of Coomassie-stained gels is notoriously difficult, did the authors use different gels for determining protein amounts vs. blotting? This would not be an appropriate normalization method for the actual blotted membranes, and would therefore need to be replaced. If these Coomassie controls were done on the same gels that went on to be blotted for antibodies, please provide technical details in the Materials and methods, as this is a non-standard approach.

The stained gels are the same gels that were transferred onto the PVDF membranes. We routinely stain all gels after wet transfer overnight at 40V. Regardless of transfer technique, there is typically protein remaining on the gel after the transfer of total cell and total tissue lysates. We find that in most cases this is a more reliable loading control than a ‘housekeeping’ protein, which can change with conditions. However, to address the concern regarding equal protein loading, we re-ran the gels in question and included pan-actin blots for the membranes in new Figure 4E (old Figure 3E) and new Figure 9B (old Figure 7B) to show that our original conclusions remain. We also modified the Materials and methods section to include information about post-transfer gel staining.

[Editors' note: further revisions were requested prior to acceptance, as described below.]

The main remaining issue is on how strong the conclusions can be drawn. Please make modifications to the text and potentially to the title to reflect the concerns of reviewer 3.Reviewer #1:The authors have added a number of new experiments to the paper that strengthen the conclusion linking phosphorylation of GFAP to caspase-6 cleavage of GFAP and to the pathology of Alexander disease. While major point 1 is not fully addressed (determining whether these changes are a cause or a consequence of more severe disease mutations), the paper does make significant new findings in the area of Alexander disease that warrant publication in eLife.

We thank reviewer #1 for their careful and positive evaluation of our revised manuscript. We have made changes to the title, Introduction, and Discussion to clarify that causality was not established with regards to the PTMs and AxD disease severity, and also in the answers to reviewer 3 comments below.

Reviewer #2:The authors have satisfactorily addressed all of my major and minor concerns; I have no further issues. Additionally, the mechanistic insight provided by the new experiments and analyses has significantly elevated the impact of the study.

We thank reviewer #2 for their careful and positive evaluation of our revised manuscript and the new data.

Reviewer #3:The authors have made a significant effort in adding important data to address the reviewers' concerns and questions. These include in vitro assembly analysis for the GFAP S13D phospho-mimic, the addition of the S13E phospho-mimic and analysis of GFAP-R79H phosphorylation sites in transfection experiments (SW13vim-), and both genetic and pharmacological inhibition of caspase-6 to assess the effects of GFAP cleavage on protein aggregation in SW13vim- cells. Important controls were also added for the iPS cell and AxD patient experiments. Most of these additions have strengthened the manuscript and support a connection between GFAP phosphorylation and aggregation.

We thank reviewer #3 for their careful evaluation of our revised manuscript and our new data further linking GFAP phosphorylation and aggregation.

However, the new data showing caspase-6 inhibition in itself disrupts GFAP filament organization and fails to rescue S13D aggregation complicate interpretation of the results, and evidence for direct cause and effect between caspase-6 GFAP cleavage and protein aggregation is still lacking. The findings in this report are important in identifying PTMs as new markers of disease severity and should be published, but as noted in the original review, the authors should be more conservative with their conclusions.

**We agree with the reviewer that additional experiments would be required to conclusively establish if there is a direct cause and effect between GFAP caspase-6 cleavage and aggregation in AxD patient cells. We modified the Introduction and Discussion to reflect that point, as noted in the responses to the specific comments below. We also agree that the PTMs highlighted in our report represent new markers of AxD severity, which is now reflected in the revised title of the manuscript.**

The authors indicate a direct link between GFAP-pSer13, caspase-6 cleavage, and protein aggregation in the title, Introduction and Discussion. A major question is still whether the observed phenomena contribute to disease severity or are downstream in the stress response and simply reflect the more severe pathology in early onset AxD.

We agree with the reviewer that more work is needed to establish where in the disease process these PTMs occur. To address this concern, we revised the statements indicating a direct link and made the following additional changes:

1) Title was changed to: **“Site-specific phosphorylation and caspase cleavage of GFAP are new markers of Alexander Disease severity”**

2) We added the following statement to the Introduction: “Collectively, our results reveal a new PTM signature that is associated with defective GFAP proteostasis in the most severe form of AxD. Future interventional studies targeting these PTMs will determine whether they contribute to, or are the consequence of, disease severity.”

3) We added the following statement to the Discussion: “While the N-terminal caspase-6 fragment of GFAP promotes filament aggregation in vitro (Chen et al., 2013), presently we do not have direct evidence of cause and effect between caspase-6 cleavage and GFAP aggregation in AxD patient cells.”

The following 3 areas contribute to this concern:1) In agreement with a more severe phenotype and astrocyte response, GFAP levels are higher in early onset AxD cases compared to those with late onset (Figure 9D). GFAP has been shown to be phosphorylated in reactive gliosis associated with certain injury models (Takemura et al., 2002), and the overall damage response could have a broader effect in activating caspase-6; p53 for example is activated in AxD (Wang et al., 2015) and is known to transactivate caspase-6 (MacLachlan and El-Deiry, 2002). The authors hypothesize that GFAP phosphorylation directly affects caspase-6 accessibility, but aggregation of GFAP-S13D and S13E in SW13vim- cells likely causes a cellular stress response that could also indirectly activate caspase-6. While I agree with the authors that investigating these mechanisms is beyond the scope of the current manuscript, they should address this possibility in their discussion and conclusions.

We agree that the PTMs could be upstream or downstream of the stress response, and have revised the text to reflect this point as follows:

1) We added the following statement to the Introduction: “Finally, we demonstrate a correlation between site-specific GFAP phosphorylation and caspase cleavage in cells and in post-mortem brain tissue from AxD patients. Although our study does not establish a causal relationship between GFAP phosphorylation and caspase cleavage, we show that caspase-6 is a new marker for the most severe form of human AxD.”

2) We added the following paragraph to the Discussion: “Our study does not address whether Ser13 phosphorylation directly promotes caspase cleavage of GFAP, or if these two PTMs are independent markers of an increased cellular stress response in AxD. One possibility is that Ser13 phosphorylation destabilizes the filament structure, thereby promoting access of caspase-6 to the rod domain Asp225 residue, where the cleavage occurs. Another likely possibility is that the increased cleavage of GFAP is an indirect result of stress-dependent caspase-6 activation in the more severe form of AxD. This is supported by previous studies showing that AxD mutations promote activation and nuclear accumulation of p53 (Wang et al., 2015), which can directly induce caspase-6 expression (MacLachlan and El-Deiry, 2002). Future studies in AxD iPSC-astrocytes and animal models will be required to determine the timing of GFAP phosphorylation and caspase-6 activation in relationship to GFAP cleavage and aggregation.”

2) New experiments to inhibit caspase-6 cleavage of GFAP produce ambiguous results. Inhibition of caspase-6 alone leads to GFAP bundling in SW13vim- cells and in combination with the S13D phospho-mimic, aggregates are not resolved. Western analysis shows clearance of a 100 kD (hmm) band, but a ~200 kD band remains and no western data is shown to demonstrate the level of WT-GFAP oligomerization induced by caspase-6 inhibition. As the authors suggest in the discussion, the role of caspase-6 activation is complex and may also be involved in cytoskeletal reorganization.

We agree that the role of caspase-6 is complex and requires thorough investigation. With respect to the hmm GFAP bands, we focused primarily on the 100kDa band in the S13D mutant because of the significant difference in the intensity of this band compared to WT GFAP (Figure 7A), and after caspase-6 inhibition (Figure 8F). We hypothesize that the 200kDa band likely reflects presence of normal GFAP tetramer, but we have not tested that specifically.

We also added the following statement to the Discussion:“However, we were not able to demonstrate in cells that inhibition of caspase-6, or mutagenesis of the cleavage site on GFAP, can resolve aggregate formation. These results point to a more complex function for caspase-6, likely involving cytoskeletal remodeling in response to stress.”

3) The supplemental data added to Figure 7 introduce important evidence for a link between GFAP phosphorylation, aggregation and caspase-6 cleavage. Immunoblotting shows de-phospho-mimics of serine residues (S to A) within the S13-S17 motif of mutant R79H GFAP eliminate high molecular weight oligomerization and significantly reduce caspase-6 cleavage fragments. However, these data are complicated by the limited contribution of pSer13 in GFAP-R79H aggregation in SW13vim- cells and the more dominant effects of pS16 and pS17, which have not been otherwise explored in this report. Although the SW13vim- cell line is commonly used to test the effects of GFAP mutation, these results demonstrate the limitations of the system for these experiments. Future work should include similar experiments in the iPSC derived astrocytes.

We agree that the limitations of the various systems need to be taken into account, and have emphasized this point in the Discussion: “This remains to be addressed in the future using the appropriate model systems, as over-expression studies in cancer cell lines (such as the SW13vim- cells we used here) may not be truly reflective of the signaling that occurs in astrocytes. In particular, it remains to be resolved whether phosphorylation of GFAP on Ser13 is part of a sequentially priming phosphorylation cascade involving nearby Ser16/17 (as predicted by the kinase motif analysis) or if Ser16/17 phosphorylation is unique to the SW13 over-expression system.”